



# Identification of lower-order inositol phosphates (IP$_5$ and IP$_4$) in soil extracts as determined by hypobromite oxidation and solution $^{31}$P NMR spectroscopy

Jolanda E. Reusser[1], René Verel[2], Daniel Zindel[2], Emmanuel Frossard[1] and Timothy I. McLaren[1]

[1]Department of Environmental Systems Science, ETH Zurich, Lindau, 8325, Switzerland
[2]Department of Chemistry and Applied Biosciences, ETH Zurich, Zurich, 8093, Switzerland

*Correspondence to*: Jolanda E. Reusser (jolanda.reusser@usys.ethz.ch)

**Abstract.** Inositol phosphates (IP) are a major pool of identifiable organic phosphorus (P) in soil. However, insight on their distribution and cycling in soil remains limited, particularly of lower-order IP (IP$_5$ and IP$_4$). This is because their quantification typically requires a series of chemical extractions, including hypobromite oxidation to isolate IP, followed by chromatographic separation. Here, for the first time, we identify the chemical nature of organic P in four soil extracts following hypobromite oxidation using solution $^{31}$P NMR spectroscopy and transverse relaxation (T$_2$) experiments. Soil samples analysed include the A horizon of a Ferralsol from Colombia, of a Cambisol from Switzerland, of a Gleysol from Switzerland and of a Cambisol from Germany. Solution $^{31}$P NMR spectra of the phosphomonoester region on soil extracts following hypobromite oxidation revealed an increase in the number of sharp signals (up to 70), and an on average 2-fold decrease in the concentration of the broad signal compared to the untreated soil extracts. We identified the presence of four stereoisomers of IP$_6$, four stereoisomers of IP$_5$, and *scyllo*-IP$_4$ (using solution $^{31}$P NMR spectroscopy). We also identified for the first time two isomers of *myo*-IP$_5$ in soil extracts: *myo*-(1,2,4,5,6)-IP$_5$ and *myo*-(1,3,4,5,6)-IP$_5$. Concentrations of total IP ranged from 1.4 to 159.3 mg P/kg$_{soil}$ across all soils, of which between 9 % and 50 % were comprised of lower-order IP. Furthermore, we found that the T$_2$ times, which are considered to be inversely related to the tumbling of a molecule in solution and hence its molecular size, were significantly shorter for the underlying broad signal compared to the sharp signals (IP$_6$) in soil extracts following hypobromite oxidation. In summary, we demonstrate the presence of a plethora of organic P compounds in soil extracts, largely attributed to IP of various order, and provide new insight on the chemical stability of complex forms of organic P associated with soil organic matter.

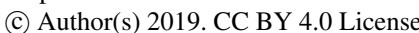



## 1 Introduction

Inositol phosphates (IP) are found widely in nature and are important for cellular function in living organisms.
They are found in eukaryotic cells where they operate in ion-regulation processes, as signalling or P storage
compounds (Irvine and Schell, 2001). The basic structure of IP consists of a carbon ring (cyclohexanehexol) with
one to six phosphorylated centers ($IP_{1-6}$) and up to nine stereoisomers (Cosgrove and Irving, 1980; Angyal, 1963).
An important IP found in nature is $myo$-$IP_6$, which is used as a P storage compound in plant seeds. Another
important species of IP is that of $myo$-(1,3,4,5,6)-$IP_5$, which is present in most eukaryotic cells at concentrations
ranging from 15 to 50 µM (Riley Andrew et al., 2006). Species of $IP_{1-3}$ are present in phospholipids such as
phosphatidylinositol diphosphates and are an essential structural component of the cell membrane system
(Cosgrove and Irving, 1980; Strickland, 1973).
Inositol phosphates have been reported to comprise more than 50 % of total organic phosphorus ($P_{org}$) in some
soils (Cosgrove and Irving, 1980; Turner, 2007; McDowell and Stewart, 2006). Four stereoisomers of IP have
been detected in soils, with the $myo$ stereoisomer being the most abundant (56 %), followed by $scyllo$ (33 %), $neo$
and D-$chiro$ (11 %) (Turner et al., 2012; Cosgrove and Irving, 1980). The largest input of $myo$-$IP_6$ to the soil occurs
via the addition of plant seeds (Turner et al., 2002). However, the addition of $myo$-$IP_6$ to soil can also occur via
manure input because monogastric animals are incapable of digesting $myo$-$IP_6$ without the addition of phytases to
their diets (Turner et al., 2007b; Leytem and Maguire, 2007). The accumulation of $myo$-$IP_6$ in soil occurs due to
the negative charge of the deprotonated phosphate groups, which can coordinate to the charged surfaces of Fe- and
Al-(hydro)-oxides (Ognalaga et al., 1994; Anderson et al., 1974), clay minerals (Goring and Bartholomew, 1951)
and soil organic matter (SOM) (McKercher and Anderson, 1989), or form insoluble precipitates with cations (Celi
and Barberis, 2007). These processes lead to the stabilisation of IP in soil resulting in its accumulation and reduced
bioavailability (Turner et al., 2002). In contrast, the sources and mechanisms controlling the flux of $scyllo$-, $neo$-
and D-$chiro$-$IP_6$ in soil remain unknown but are thought to involve epimerization of the $myo$ stereoisomer
(L'Annunziata, 1975).
Chromatographic separation of alkaline soil extracts revealed the presence of four stereoisomers of $IP_6$ and lower-
order $IP_{1-5}$ (Halstead and Anderson, 1970; Irving and Cosgrove, 1982; Cosgrove and Irving, 1980; Anderson and
Malcolm, 1974). Irving and Cosgrove (1981) used hypobromite oxidation prior to chromatography to isolate the
IP fraction in alkaline soils. The basis of this approach is that IP are considered to be highly resistant to
hypobromite oxidation, whereas other organic compounds (e.g. phospholipids and nucleic acids) will undergo
oxidation (Dyer and Wrenshall, 1941; Turner and Richardson, 2004). The resistance of IP to hypobromite
oxidation is thought to be due to the high charge density and steric hindrance, which is caused by the chair
conformation of the molecule and the bound phosphate groups, with the P in its highest oxidation state.
Hypobromite oxidation of inositol (without phosphate groups) mainly results in the formation of inososes, which
have an intact carbon ring (Fatiadi, 1968). Fatiadi (1968) considered that the oxidation of bromine with inositol is
stereospecific and comparable to catalytic or bacterial oxidants.
A limitation of chromatographic separation of alkaline extracts is that there is a mixture of unknown organic
compounds that can co-elute with IP, and result in an overestimation of IP concentrations (Irving and Cosgrove,
1981). However, this can also occur for IP, and historically, studies often reported the combined concentration of
$IP_6$ and $IP_5$ due to a lack of differentiation in their elution times (McKercher and Anderson, 1968a). More recently,
Almeida et al. (2018) investigated how cover crops might mobilize soil IP using hypobromite oxidation on NaOH-
EDTA extracts followed by chromatographic separation. The authors found that pools of $myo$-$IP_6$ and 'unidentified



IP' accounted for 30 % of the total extractable pool of P and hypothesised that the 'unidentified IP' pool consists
solely of lower-order *myo*-IP. Pools of lower order $IP_{1-5}$ comprise on average 17 % of the total pool of IP in soil
and account for an important pool of soil organic P in terrestrial ecosystems (Turner et al., 2002; Anderson and
Malcolm, 1974; Cosgrove and Irving, 1980; Turner, 2007).
Since the 1980s, solution $^{31}$P nuclear magnetic resonance spectroscopy (NMR) has been the most commonly used
technique to characterise the chemical nature of organic P in soil extracts (Cade-Menun and Liu, 2014; Newman
and Tate, 1980). An advantage of this technique is the simultaneous detection of all forms of organic P that come
into solution, which is brought about by a single step extraction with alkali (Cade-Menun and Preston, 1996).
However, a limitation of the technique has been the loss of information on the diversity and amount of soil IP
compared to that typically obtained prior to 1980 (Cosgrove, 1963; Smith and Clark, 1951; Anderson, 1955). To
date, solution $^{31}$P NMR spectroscopy on soil extracts has only reported concentrations of *myo*-, *scyllo*-, *chiro*- and
*neo*-$IP_6$. The fact that lower-order IP were not reported in studies using NMR spectroscopy might be due to overlap
of peaks in the phosphomonoester region, which makes peak assignment of specific compounds difficult (Doolette
et al., 2009).
Turner et al. (2012) carried out hypobromite oxidation prior to solution $^{31}$P NMR analysis of alkaline soil extracts
to isolate the IP fraction. This had the advantage of reducing the number of NMR signal in the phosphomonoester
region and consequently the overlap of peaks. The authors demonstrated the presence of *neo*- and *chiro*-$IP_6$ in
NMR spectra via spiking of brominated extracts. Interestingly, the authors also reported the presence of NMR
signals in the phosphomonoester region that could not be assigned to $IP_6$ and was resistant to hypobromite
oxidation. They were not able to attribute the NMR signals to any specific P compounds, but hypothesised based
on their resistance to hypobromite oxidation that they were due to lower-order IP.
The aim of this study was to identify and quantify IP in soil extracts following hypobromite oxidation using
solution $^{31}$P NMR spectroscopy. In addition, the structural composition of phosphomonoesters in soil extracts
following hypobromite oxidation was probed using solution $^{31}$P NMR spectroscopy and transverse relaxation
experiments. We hypothesise that a large portion of sharp peaks in the phosphomonoester region of untreated soil
extracts would be resistant to hypobromite oxidation, which would indicate the presence of IP.
**2      Experimental section**
**2.1      Soil collection and preparation**
Soil samples were collected from the upper horizon of the profile at four diverse sites. These include a Ferralsol
from Colombia, a Vertisol from Australia, a Cambisol from Germany, and a Gleysol from Switzerland (FAO,
2014). Background information and some chemical properties of the soils are reported in Table 1. Briefly, the
Ferralsol was collected from an improved grassland in 1997 at the Carimagua Research Station's long-term
Culticore field experiment in Columbia (Bühler et al., 2003). The Vertisol was collected from an arable field in
2018 located in southern Queensland. The site had been under native shrubland prior to 1992. The Cambisol was
collected from a beech forest in 2014, and is part of the "SPP 1685 – Ecosystem Nutrition" project (Lang et al.,
2017; Bünemann et al., 2016). The Gleysol was collected from the peaty top soil layer of a drained marshland in
2017, which has been under grassland for at least 20 years.
Soil samples were passed through a 5 mm sieve and dried at 60°C for 5 days, except for the Ferralsol (sieved <2
mm) and the Vertisol (ground <2 mm), which were received dried. Total concentrations of C and N in soils were





obtained using combustion of 50 mg ground soil (to powder) weighed into tin foil capsules (vario PYRO cube®,
Elementar Analysesysteme GmbH). Soil pH was measured in $H_2O$ with a soil to solution ratio of 1:2.5 (w/w) using
a glass electrode.
[Suggested location Table 1]

### 2.2    Soil phosphorus analyses

Total concentrations of soil P were carried out by X-ray fluorescence spectroscopy (SPECTRO XEPOS ED-XRF,
AMETEK®) using 4.0 g of ground to powder soil sample mixed with 0.9 g of wax (CEREOX Licowax,
FLUXANA®). The XRF instrument was calibrated using commercially available reference soils. Concentrations
of organic P for NMR analysis were carried out using the NaOH-EDTA extraction technique of Cade-Menun et
al. (2002) at a soil to solution ratio of 1:10. This extraction procedure is the same as described in McLaren et al.
118    (2019).

### 2.3    Hypobromite oxidation

Hypobromite oxidation of NaOH-EDTA soil filtrates was carried out based on a modified version of the method
described in Suzumura and Kamatani (1993) and Turner et al. (2012). Briefly, 10 mL of the filtrate was placed in
a three necked round bottom flask equipped with a septum, a condenser, a magnetic stir bar and thermometer
(through a claisen adapter with $N_2$ adapter). After the addition of 1 mL 10 M aqueous NaOH and vigorous stirring,
an aliquot of 0.6 mL $Br_2$ (which was cooled prior to use) was added, resulting in an exothermic reaction where
some of the soil extracts nearly boiled. The optimal volume of $Br_2$ for oxidation was assessed in a previous pilot
study using 0.2, 0.4, 0.6 and 0.8 mL $Br_2$ volumes, and then observing differences in their NMR spectral features
(data not shown). The reaction was heated to 100 °C within 10 min and kept at reflux for an additional 5 min. After
cooling to room temperature, the solution was acidified with 2 mL of 6 M aqueous HCl solution in order to obtain
a pH < 3, which was confirmed with a pH test strip. The acidified solution was reheated to 100 °C for 5 min under
a stream of nitrogen to vaporise any excess bromine. The pH of the solution was gradually increased to 8.5 using
10 M aqueous NaOH solution. After dilution with 10 mL of $H_2O$, 5 mL 50 % (w/w) ethanol and 10 mL 10 % (w/w)
barium acetate solution was added to the solution in order to precipitate any IP (Turner et al., 2012). The solution
was then heated and boiled for 10 min and allowed to cool down overnight. The solution was subsequently
transferred to a 50 mL centrifuge tube and a 10 mL aliquot of 50 % (w/w) ethanol was added, manually shaken,
and centrifuged at 1500 g for 15 min. The supernatant was removed and a 15 mL aliquot of 50 % (w/w) ethanol
was added to the precipitate, shaken, and then centrifuged again as before. The supernatant was removed and the
process repeated once more to further purify the pool of IP. Afterwards, the precipitate was transferred with 20
mL of $H_2O$ into a 100 mL beaker that contained a 20 mL volume (equating to a mass of 15 g) of Amberlite® IR-
120 cation exchange resin beads in the $H^+$ form (Sigma-Aldrich, product no. 06428). The suspension was stirred
for 15 min and then passed through a Whatman no. 42 filter paper. A 9 mL aliquot of the filtrate was frozen at
− 80 °C and then lyophilised prior to NMR analysis. This resulted in 18 - 26 mg of lyophilised material across all
soils. Concentrations of total P in solutions were obtained using inductively coupled plasma-optical emission
spectrometry (ICP-OES). Concentrations of molybdate reactive P (MRP) were obtained using the malachite green
method of Ohno and Zibilske (1991). The difference in concentrations of total P and MRP in solution is that of
molybdate unreactive P (MUP), which is considered to be largely that of organic P.



To assess the of effect hypobromite oxidation on the stability of an $IP_6$, a duplicate sample of the Cambisol and
the Gleysol was spiked with 0.1 mL of a 11 mM $myo$-$IP_6$ standard. The recovery of the added $myo$-$IP_6$ following
hypobromite oxidation was calculated using Eq. (1):
$$Spike\ recovery\ (\%) = \frac{C_{spiked}\left(\frac{mg}{L}\right) - C_{unspiked}\left(\frac{mg}{L}\right)}{C_{standard\ added}\left(\frac{mg}{L}\right)},$$  (1)
where $C_{spiked}$ and $C_{unspiked}$ are the concentrations of $myo$-$IP_6$ in NaOH-EDTA extracts following hypobromite
oxidation of the spiked and unspiked samples, respectively. $C_{standard\ added}$ is the concentration of the added $myo$-$IP_6$
within the standard. As $^{31}P$ NMR spectroscopy of the standard revealed impurities, the concentration of $myo$-$IP_6$
in the standard was calculated based on the $^{31}P$ NMR spectrum.

### 2.4 Sample preparation for solution $^{31}P$ NMR spectroscopy

The lyophilised material of the untreated soil extracts was prepared for solution $^{31}P$ NMR spectroscopy based on
a modification of the methods of Vincent et al. (2013) and Spain et al. (2018). Briefly, 120 mg of lyophilised
material was taken and dissolved in 600 μL of 0.25 M NaOH-0.05 M $Na_2$EDTA solution (ratio of 1:5). However,
for the Cambisol sample, this ratio resulted in a NMR spectrum that exhibited significant line broadening.
Therefore, this was repeated on a duplicate sample but at a smaller lyophilised material to solution ratio (ratio of
1:7.5), as suggested in Cade-Menun and Liu (2014), which resolved the issue of poor spectral quality. The
suspension was stored overnight to allow for complete hydrolysis of phospholipids and RNA (Doolette et al., 2009;
Vestergren et al., 2012), which was then centrifuged at 10621 g for 15 min. A 500 μL aliquot of the supernatant
was taken, which was subsequently spiked with a 25 μL aliquot of a 0.03 M methylenediphosphonic acid standard
made in $D_2O$ (Sigma-Aldrich, product no. M9508) and a 25 μL aliquot of sodium deuteroxide at 40 % (w/w) in
$D_2O$ (Sigma-Aldrich, product no. 372072). The solution was then mixed and transferred to a 5 mm diameter NMR
tube.
A similar procedure was used for the soil extracts that had undergone hypobromite oxidation, except the total mass
of lyophilised material (18 - 26 mg) was dissolved with 600 μL of a 0.25 M NaOH-0.05 M $Na_2$EDTA solution.
However, for the Cambisol sample, the NMR spectrum exhibited considerable line-broadening, and an additional
400 μL aliquot of NaOH-EDTA solution was added to the NMR tube, mixed, and then returned to the NMR
spectrometer. This resolved the issue of poor spectral quality.

### 2.5 Solution $^{31}P$ NMR spectroscopy

Solution $^{31}P$ NMR analyses were carried out on all untreated and hypobromite oxidised soil extracts at the NMR
facility of the Laboratory of Inorganic Chemistry (Hönggerberg, ETH Zürich). All spectra were obtained with a
Bruker AVANCE III MD 500 MHz NMR spectrometer equipped with a cryogenic probe (CryoProbe™ Prodigy)
(Bruker Corporation; Billerica, MA). The $^{31}P$ frequency for this NMR spectrometer was 202.5 MHz and gated
broadband proton decoupling with a 90° pulse of 12 μs was applied. Spectral resolution under these conditions for
$^{31}P$ was < 1 Hz. Longitudinal relaxation ($T_1$) times were determined for each sample with an inversion recovery
experiment (Vold et al., 1968). This resulted in recycle delays ranging from 8.7 to 30.0 sec for the untreated
extracts and 7.8 to 38.0 sec for the hypobromite oxidised soil extracts. The number of scans for the untreated
extracts was set to 1024 or 4096, depending on the signal to noise ratio of the obtained spectrum. All hypobromite
oxidised spectra were acquired with 3700 to 4096 scans.


**2.6      Processing of NMR spectra**
All NMR spectra were processed with Fourier transformation, phase correction, and baseline adjustment within
the TopSpin® software environment (Version 3.5 pl 7, Bruker Corporation; Billerica, MA). Line broadening was
set to 0.6 Hz. Quantification of NMR signals involved obtaining the integrals of the following regions: 1) up to
four phosphonates ($\delta$ 19.8 to 16.4 ppm); 2) the added MDP ($\delta$ 17.0 to 15.8ppm) including its two carbon satellite
peaks; 3) the combined orthophosphate and phosphomonoester region ($\delta$ 6.0 to 3.0 ppm); 4) up to four
phosphodiesters ($\delta$ 2.5 to -3.0 ppm), and 5) pyrophosphate ($\delta$ -4.8 to -5.4 ppm). Due to overlapping peaks in the
orthophosphate and phosphomonoester region, spectral deconvolution fitting was applied as described in McLaren
et al. (2019). The NMR observability of total P ($P_{tot}$) in NaOH-EDTA extracts was calculated using Eq. (2)
(Doolette et al., 2011b; Dougherty et al., 2005):
$$NMR\ observability\ (\%) = \frac{P_{tot}\ NMR}{P_{tot}\ ICP-OES} * 100\ \%\ ,\qquad (2)$$
where $P_{tot}$ NMR refers to the total P content detected in the soil extracts using solution $^{31}$P NMR spectroscopy and
$P_{tot}$ ICP-OES refers to the total P concentration measured in the soil extracts prior to freeze-drying using ICP-OES.
**2.7      Spiking experiments**
To identify the presence of IP in hypobromite oxidised extracts, samples were spiked with a range of standards
and then analysed again using NMR spectroscopy. This involved the addition of 5 to 20 μL aliquots of an IP
standard solution directly into the NMR tube, which was then sealed with parafilm, manually shaken, and then
allowed to settle prior to NMR analysis. Each sample extract was consecutively spiked with no more than four IP
standards. The NMR spectra of soil extracts after spiking were overlaid with the NMR spectra of unspiked soil
extracts to identify the presence of IP across all soil samples. This comparison of NMR spectra was possible due
to negligible changes in the chemical shifts of peaks among soil samples. The IP standards used in this study are
listed in Table 2.
[Suggested location Table 2]
**2.8      Transverse relaxation (T$_2$) experiments**
Due to the presence of sharp and broad signals in the phosphomonoester region of NMR spectra on hypobromite
oxidised soil extracts, transverse relaxation (T$_2$) experiments were carried out to probe their structural composition.
The transverse relaxation (originally spin-spin relaxation) describes the loss of magnetisation in the x-y plane. This
loss occurs due to magnetic field differences in the sample, arising either by instrumentally caused magnetic field
inhomogeneities or by local magnetic fields in the sample caused by intramolecular and intermolecular interactions
(Claridge, 2016). Generally, small, rapidly tumbling molecules exhibit longer T$_2$ relaxation times compared to
large, slowly tumbling molecules (McLaren et al., 2019).
Briefly, solution $^{31}$P NMR spectroscopy with a Carr-Purcell-Meiboom-Gill (CPMG) pulse sequence (Meiboom
and Gill, 1958) was carried out on all hypobromite oxidised soil extracts, as described in McLaren et al. (2019).
This involved a constant spin-echo delay ($\tau$) of 5 ms, which was repeated for a total of eight iterations (spin-echo
periods of 5, 50, 100, 150, 200, 250, 300, and 400 ms). A total of 4096 scans and a recycle delay of 4.75 sec was
used for all iterations. Transverse relaxation times for the aforementioned integral ranges were calculated using
Eq. (3) within the TopSpin® software environment. Due to overlapping peaks in the orthophosphate and
phosphomonoester region, spectral deconvolution was carried out to partition the NMR signal, as described in



McLaren et al. (2019). The $T_2$ times of the partitioned NMR signals were calculated using Eq. (3) within RStudio©
(version 1.1.442):
$$M(t) = M_0 * e^{(-t*T_2^{-1})} ,$$   (3)
where M refers to the net magnetisation derived from the average angular momentum in the x–y plane, τ refers to
the spin-echo delay in milliseconds (ms), and $T_2$ refers to the transverse relaxation time (ms).
**2.9    Statistical analyses and graphics**
Statistical analyses were carried out using Microsoft® Excel 2016 and MATLAB R2017a (©The MathWorks,
Inc.). Graphics were created with Microsoft® Excel 2016 and MATLAB R2017a (©The MathWorks, Inc.).
Solution (1D) $^{31}$P NMR spectra were normalised to the peak intensity of MDP (δ 16.46 ppm). Spectra from the $T_2$
experiments were normalised to the peak intensity of *scyllo*-IP$_6$ (δ 3.22 ppm).
A one-way ANOVA was carried out in MATLAB R2017a (©The MathWorks, Inc.) with a subsequent multi
comparison of mean values using the Tukey's honestly significant difference procedure based on the studentised
range distribution (Hochberg and Tamhane, 1987; Milliken and Johnson, 2009).
**3    Results**
**3.1    Phosphorus concentrations in soil extracts**
Concentrations of total soil P as determined by XRF ranged from 320 to 3841 mg P/kg$_{soil}$ across all soils (Table
3). Concentrations of total P as estimated by the NaOH-EDTA extraction technique ranged from 160 to
1850 mg P/kg$_{soil}$, which comprised 28 to 51 % of the total soil P as determined by XRF. Pools of organic P
comprised 28 to 72 % of the total P in NaOH-EDTA untreated soil extracts.
Concentrations of total P in NaOH-EDTA soil extracts following hypobromite oxidation ranged from 77 to 578 mg
P/kg$_{soil}$ (Table 3), which accounted for 31 to 48 % (on average 38 %) of the total P originally present in the extracts.
Similarly, pools of organic P in NaOH-EDTA extracts following hypobromite oxidation were lower, comprising
22 to 48 % (on average 36 %) of that originally present in untreated NaOH-EDTA extracts across all soils.
[Suggested location Table 3]
**3.2    Solution $^{31}$P NMR spectra of hypobromite oxidised soil extracts**
The most prominent signal in the NMR spectra of untreated NaOH-EDTA soil extracts was that of orthophosphate
at δ 5.25 (±0.25) ppm, followed by the phosphomonoester region ranging from δ 6.0 to 3.0 ppm (Fig. 1). There
were also some minor signals due to pyrophosphate δ -5.06 (±0.19) ppm (all soils), phosphodiesters ranging from
δ 2.5 to -2.4 ppm (not detected in the Vertisol), and phosphonates (not including the added MDP) at δ 19.8, 19.2
and 18.3 ppm (not detected in the Gleysol). However, these compounds comprised less than 8 % of the total NMR
signal.
Following hypobromite oxidation of NaOH-EDTA extracts, the most prominent NMR signals were found in the
orthophosphate (65 % of total NMR signal) and phosphomonoester (35 % of total NMR signal) region across all
soils (Fig. 1). Phosphodiesters and pyrophosphates were removed following hypobromite oxidation in the
Ferralsol, the Vertisol and the Cambisol (DE). Although, some signal remained in the Gleysol at low
concentrations (0.4 % of the total NMR signal). Phosphonates were removed following hypobromite oxidation in



the Ferralsol and the Vertisol, but a total of five sharp peaks in the phosphonate region were detected (δ 19.59,
18.58, 17.27 and 9.25 ppm) in the Cambisol. These peaks comprised 0.6 % of the total NMR signal.
The phosphomonoester region of NMR spectra on untreated NaOH-EDTA extracts exhibited two main features:
1) the presence of a broad signal centered at around δ 4.1 (±0.1) ppm with an average line-width at half height of
256.12 Hz; and 2) the presence of between 19 and 34 sharp signals. This was similarly the case on hypobromite
oxidised extracts, except there was a decrease in the intensity of the broad signal and a change in the distribution
and intensity of sharp signals. For the Cambisol and Gleysol, the number of sharp signals in the phosphomonoester
region approximately doubled (to 40 and 70 sharp signals, respectively) following hypobromite oxidation. In
contrast, less than half of the sharp signals remained in the Ferralsol following hypobromite oxidation (i.e. 14 of
the 30 peaks originally present in the untreated extract), whereas one peak was removed following hypobromite
oxidation in the Vertisol. There was little change (0.23 ppm) in the chemical shifts of peaks between the untreated
and hypobromite oxidised extracts.






**Figure 1. Solution $^{31}$P nuclear magnetic resonance (NMR) spectra (500 MHz) of the orthophosphate and phosphomonoester region on untreated (UT) and hypobromite oxidised (HO) 0.25 M NaOH + 0.05 M EDTA soil extracts (Ferralsol, Vertisol, Cambisol and Gleysol). Signal intensities were normalised to the MDP peak intensity. The vertical axes were increased for improved visibility of spectral features, as indicated by a factor. The orthophosphate peak is marked with an asterisk. The symbol '×' marks the four individual peaks of *myo*-IP$_6$ and '+' the peak of *scyllo*-IP$_6$.**





[Suggested location Table 4]

**3.3 Identification and quantification of inositol phosphates (IP$_6$, IP$_5$ and IP$_4$) in soil extracts**

A detailed view of the phosphomonoester region of spiked extracts is shown in Fig. SI1 to SI5 of the Supporting
Information. The number of identified sharp peaks in the phosphomonoester region ranged from 7 (Vertisol) to 33
(Gleysol). *myo*- and *scyllo*-IP$_6$ were identified in the hypobromite oxidised extracts of all soils (Table 5). On
average, 72 % of *myo*-IP$_6$ and 56 % of *scyllo*-IP$_6$ present in the untreated extracts remained in the hypobromite
oxidised extracts (Table SI1 in the Supporting Information). *neo*-IP$_6$ was identified in the the 2-equatorial/4-axial
and 4-equatorial/2-axial conformations, and c*hiro*-IP$_6$ in the 2-equatorial/4-axial confirmation, of the oxidised
extracts in the Cambisol and Gleysol, but were absent in the Ferralsol and the Vertisol (Fig. SI4 and SI5 in the
Supporting Information).
The *myo*, *scyllo*, *chiro* and *neo* stereoisomers of IP$_5$ were identified in various hypobromite oxidised extracts (Table
5). Two isomers of *myo*-IP$_5$ were identified in some extracts, which included *myo*-(1,2,4,5,6)-IP$_5$ and *myo*-
(1,3,4,5,6)-IP$_5$. In addition, *scyllo*-IP$_4$ was detected in all soils except that of the Vertisol. There was insufficient
evidence for the presence of *myo*-IP$_4$ in these soil samples, as only one of the two peaks of this compound was
present in the NMR spectra of untreated extracts.
Concentrations of total IP ranged from 1.4 to 159.3 mg P/kg$_{soil}$ across all soils, which comprised between 1 %
(Vertisol) and 18 % (Gleysol) of the organic P in untreated NaOH-EDTA extracts (Table 3). Pools of IP$_6$ were the
most abundant form of IP, which ranged from 0.9 to 144.8 mg P/kg$_{soil}$ across all soils (Table 5). The proportion of
IP$_6$ stereoisomers across all soils were in the order of *myo* (61 %, SD=12), *scyllo* (29 %, SD=3), *chiro* (6 %, SD=8)
and *neo* (4 %, SD=5). Similarly, the *myo* and *scyllo* stereoisomer were also the most predominant forms of IP$_5$,
but comprised between 83 % (Cambisol) and 100 % (Ferralsol and Vertisol) of total IP$_5$ (Table 5). Trace amounts
of *scyllo*-IP$_4$ were also detected in three of the four soils. The ratio of total IP$_6$ to IP$_5$ differed across all soils (Fig.

293 2).

[Suggested location Table 5]





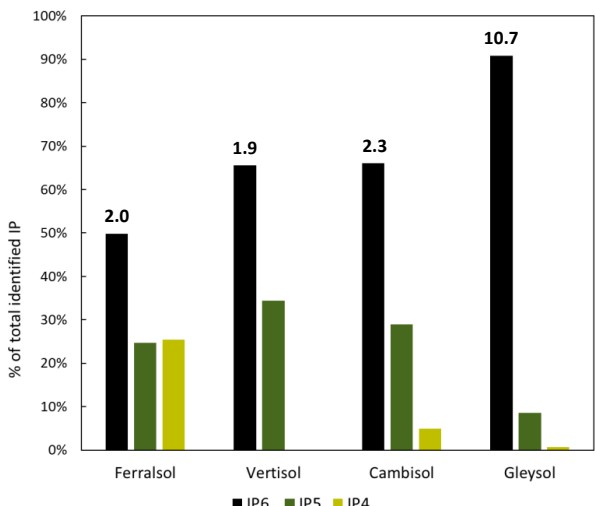

**Figure 2. The proportion of total identifiable pools of inositol hexakisphosphates (IP₆), -pentakisphosphates (IP₅) or – tetrakisphosphates (IP₄) to that of the total pool of identifiable IP, as determined by solution ³¹P NMR spectroscopy on four soil extracts (Ferralsol, Vertisol, Cambisol and Gleysol) following hypobromite oxidation. Values located above the IP₆ bar are the ratio of total identifiable IP₆ to that of IP₅ in each soil sample.**


If sharp peaks arising from IP were identified in the NMR spectra on hypobromite oxidised extracts, a comparison
was made with that of their corresponding untreated extracts. The sharp peaks of all stereoisomers of $IP_6$ were
present in the untreated extracts. The five peaks of *myo*-(1,2,4,5,6)-$IP_5$ and the three peaks of *scyllo*-$IP_5$ were also
identified. However, it was not possible to clearly identify other $IP_5$ compounds in untreated extracts due to
overlapping signals. In the Gleysol, all three peaks of *scyllo*-$IP_5$ were detected, but only two of the possible five
peaks could be clearly assigned to *myo*-(1,2,4,5,6)-$IP_5$. In the Ferralsol, both peaks of *scyllo*-$IP_4$ were present in
the untreated extract, but only two of the three possible peaks could be assigned to *scyllo*-$IP_5$. In the Vertisol, no
$IP_5$ was identified. Concentrations of IP in untreated extracts assessed by spectral deconvolution fitting were
generally double than that measured in hypobromite oxidised extracts. Recoveries of added *myo*-$IP_6$ in the Gleysol
and Cambisol following hypobromite oxidation were 47 % and 20 %, respectively.
**3.4      Spin-echo analysis of selected P compounds**
Due to the presence of sharp and broad signals in hypobromite oxidised soil extracts, the structural composition
of phosphomonoesters was probed. A comparison of the NMR spectra at the lowest ($1*\tau$) and highest ($80*\tau$) pulse
delays revealed a fast decaying broad signal for all hypobromite oxidised soil extracts, which was particularly
evident in the Gleysol (Fig. 3). Calculated $T_2$ times of all $IP_6$ stereoisomers were longer than that of the broad
signal (Table 6). The $T_2$ times of *scyllo*-IP6 (on average 175.8 ms, SD=49.7) were generally the longest of all
stereoisomers of $IP_6$. The $T_2$ time of the orthophosphate peak was the shortest, which was on average 11.5 ms
(SD=4.9).
The average (n=4) $T_2$ times of the broad peak was significantly different than that of *scyllo*- and *myo*-$IP_6$ ($p <$
0.05). Significant differences in the $T_2$-times of *neo*- and D-*chiro*-$IP_6$ were not tested, as these compounds were
not detected in the Ferralsol and the Vertisol.
[Suggested location Table 6]

Figure 3. Solution $^{31}$P NMR spectra of hypobromite oxidised soil extracts acquired with a CPMG pulse sequence with **1\*τ (black) and 80\*τ (red) spin-echo delays. The orthophosphate (\*), *scyllo*-IP$_6$ (+) and *myo*-IP$_6$ peaks (×) are marked accordingly. Spectra were normalised to the maximum *scyllo*-IP$_6$ peak intensity in the 1\*τ spectrum for each soil. The vertical axes were increased/decreased for better visualisation by an indicated factor.**

## 4       Discussion

### 4.1     Pools of phosphorus in untreated and hypobromite oxidised soil extracts

On average, 44 % of total P (as measured with XRF) was extracted by NaOH-EDTA, which is consistent with
previous studies (McLaren et al., 2019; Li et al., 2018; Turner, 2008). The non-extractable pool of P is likely to
comprise of inorganic P as part of insoluble mineral phases, but could also contain some organic P (McLaren et
al., 2015a). Nevertheless, the NaOH-EDTA extraction technique is considered to be a measure of total organic P
in soil, which can be subsequently characterised by solution $^{31}$P NMR spectroscopy (Cade-Menun and Preston,

326     1996).

Hypobromite oxidation resulted in a decrease in the concentration of inorganic and organic P in NaOH-EDTA
extracts across all soils. The decrease of organic P is consistent with previous studies (Turner et al., 2012; Almeida
et al., 2018; Turner and Richardson, 2004). However, Almeida et al. (2018) reported an overall increase in the
concentration of inorganic P following hypobromite oxidation, which the authors proposed to be caused by the
degradation of organic P forms not resistant to hypobromite oxidation. A decrease in the concentration of organic
P in NaOH-EDTA extracts following hypobromite oxidation was expected based on the oxidation of organic
molecules containing P. This will result in the production of carbon dioxide and simple organic acids (Irving and
Cosgrove, 1981; Sharma, 2013).
Overall, hypobromite oxidation of NaOH-EDTA soil extracts resulted in a considerable increase in the number of
sharp peaks and a decrease in the broad underlying peak in the phosphomonoester region compared to that of
untreated soil extracts. This was particularly the case for the Cambisol and the Gleysol, which had high
concentrations of extractable organic P. Since the broad peak is thought to be closely associated with the SOM
(Bünemann et al., 2008; Dougherty et al., 2007; McLaren et al., 2015b), its decrease in soil extracts following
hypobromite oxidation is consistent with that observed for other organic compounds (Turner et al., 2012). Our
results indicate that the majority of sharp peaks present in the phosphomonoester region of untreated soil extracts
are stable to hypobromite oxidation, and are therefore likely to be IP.
Across all soils, 5 to 15 peaks in the phosphomonoester region were removed following hypobromite oxidation
compared to those in untreated extracts, which are likely due to the oxidation of: α-and β-glycerophosphate
(Doolette et al., 2009; McLaren et al., 2015b), RNA mononucleotides (8 peaks) (Vincent et al., 2013), glucose 6-
phosphate, phosphocholine, glucose 1-phosphate, or phosphorylethanolamine (Cade-Menun, 2015).

**4.2      Phosphorus assignments of sharp peaks in hypobromite oxidised extracts**

The detection of *myo-, scyllo-, chiro*, and *neo*-IP$_6$ in untreated and hypobromite oxidised soil extracts is consistent
with previous studies using chromatography (Irving and Cosgrove, 1982; Almeida et al., 2018) and NMR (Turner
and Richardson, 2004; McLaren et al., 2015b; Jarosch et al., 2015; Vincent et al., 2013; Doolette et al., 2011a).
Turner et al. (2012) suggested that hypobromite oxidised extracts only contained *neo*-IP$_6$ in the 4-equatorial/2-
axial conformation due to the absence of signals from the 2-equatorial/4-axial conformation. In the current study,
both conformations could be identified, which is likely due to improved spectral resolution and sensitivity. The
relative abundances of the four identified stereoisomers of IP$_6$ in soil extracts were similar to previous studies
(Turner et al., 2012; Irving and Cosgrove, 1982).
Several studies have shown overlap of peaks relating to RNA mononucleotides and that of α-and β-
glycerophosphate, which are the alkaline hydrolysis products of RNA and phospholipids, respectively. However,
in the current study, several sharp peaks were present in hypobromite oxidised extracts which are in the chemical
shift range of RNA mononucleotides and α-and β-glycerophosphate. Whilst a peak at δ 4.36 ppm would be
assigned to α-glycerophosphate based on spiking experiments in the untreated extracts of the Cambisol and the
Gleysol (Doolette et al., 2009), hypobromite oxidation revealed the presence of the 2-equatorial/4-axial C2,5 peak
of *neo*-IP$_6$ at δ 4.37 ppm, and also an unidentified peak at δ 4.36 ppm in the Cambisol. Therefore, the assignment
and concentration of α-glycerophosphate may be unreliable in some soils of previous studies.
For the first time, we identified lower-order IP (IP$_5$ and IP$_4$) in soil extracts using solution $^{31}$P NMR spectroscopy.
Smith and Clark (1951) were the first to suggest the presence of IP$_5$ in soil extracts using anion-exchange
chromatography, which was later confirmed (Cosgrove, 1963; McKercher and Anderson, 1968a; Anderson, 1955).
Halstead and Anderson (1970) reported the presence of all four stereoisomers (*myo, scyllo, neo* and *chiro*) in the
lower ester fractions (IP$_2$-IP$_4$) as well as the higher ester fractions (IP$_5$, IP$_6$) isolated from soil, with the *myo*
stereoisomer being the main form in all fractions. In the current study, all four stereoisomers of IP$_5$ could be
detected in the hypobromite oxidised soil extracts, which the *myo* and *scyllo* stereoisomers were the most abundant.



The relative abundances of IP$_5$ stereoisomers are consistent with the findings of Irving and Cosgrove (1982) using
gas-liquid chromatography on the combined IP$_6$ + IP$_5$ fraction. The detection of all four stereoisomers of IP$_5$ in
NMR spectra provides direct spectroscopic evidence for their existence in soil extracts.
In addition to the four stereoisomers of IP$_5$, we were able to identify the presence of two isomers of *myo*-IP$_5$ in the
Cambisol and Gleysol, i.e. *myo*-(1,2,4,5,6)-IP$_5$ and *myo*-(1,3,4,5,6)-IP$_5$. These two isomers have not yet been
detected in soil extracts. A distinction of different *myo*-IP$_5$ isomers was not reported in earlier studies using
chromatographic separation. In non-soil extracts, *myo*-(1,2,4,5,6)-IP$_5$ was detected by Doolette and Smernik
(2016) in grapevine canes, and *myo*-(1,3,4,5,6)-IP$_5$ as the thermal decomposition product of a phytate standard
(Doolette and Smernik, 2018). Sun et al. (2017) reported *myo*-(1,2,4,5,6)-IP$_5$ as an intermediate in the major
pathways of Aspergillus niger phytase and acid phosphatase (potato) phytate degradation. The presence of *myo*-
(1,2,3,4,6)-IP$_5$ could not be confirmed as NMR analyses on the compound itself exhibited a broad NMR signal
(Fig. SI7 in the Supporting Information). This is because in solutions with a pH of 9.5 or above, the
1axial/5equatorial and 5axial/1 equatorial forms of *myo*-(1,2,3,4,6)-IP$_5$ are in a dynamic equilibrium, which can
cause broadening (Volkmann et al., 2002). According to Turner and Richardson (2004) and Chung et al. (1999),
the two identified *scyllo*-IP$_4$ peaks (signal pattern 2:2) can be attributed to the *scyllo*-(1,2,3,4)-IP$_4$ isomer. However,
the two peaks of *scyllo*-IP$_4$ overlapped in the Cambisol and Gleysol with the peak at the furthest upfield chemical
shift of *myo*-(1,2,4,5,6)-IP$_5$ at δ 3.25 ppm, and with the peak at the furthest downfield chemical shift of *myo*-
(1,3,4,5,6)-IP$_5$ at δ 4.12 ppm.
Whilst on average 48 % of the sharp peaks in the phosphomonoester region of soil extracts following hypobromite
oxidation could be attributed to IP$_6$, IP$_5$ and *scyllo*-IP$_4$, the identity of many sharp peaks remain unknown. An
unidentified peak at δ 4.33 ppm is present in all soil samples except in the Ferralsol, with concentrations of up to
10 mg P/kg$_{soil}$ (Cambisol). Other unidentified peaks at δ 3.49, 3.86, 4.20 and 3.91 ppm were detected in all soils,
with concentrations ranging from 1 to 2 mg P/kg$_{soil}$. Interestingly, two peaks upfield of *scyllo*-IP$_6$ became more
prominent (at δ 3.08, 3.05 ppm) following hypobromite oxidation, which was particularly the case in the Vertisol
soil. The diversity of organic P species in the Vertisol soil appears to be much greater than previously reported
(McLaren et al., 2014). We hypothesise that many of these unidentified peaks arise from other isomers of *myo*-
and *scyllo*-IP$_5$, based on the higher abundance of their IP$_6$ counterparts.
The ratio of IP$_6$ to lower-order IP varied across soils, which ranged in decreasing order: Gleysol ≫ Cambisol >
Vertisol > Ferralsol. McKercher and Anderson (1968b) found a higher ratio of IP$_6$ to IP$_5$ in some Scottish soils
(ratio 1.8 to 4.6) compared to some Canadian soils (0.9 to 2.4). The authors attributed this difference to the greater
stabilization of IP$_6$ relative to lower esters in the Scottish soils, possible due to climatic reasons or effects of
different soil properties. In a subsequent study, McKercher and Anderson (1968a) observed increased IP contents
with increasing total organic P content. Studies of organic P speciation along chronosequences found that *myo*-IP$_6$
concentrations declined in older soils (Turner et al., 2007a; McDowell et al., 2007). Similarly, Baker (1976) found
that the IP$_6$ + IP$_5$ concentrations in the Franz Josef chronosequence increased until 1000 years, followed by a rapid
decline. In our soil samples, the highest IP$_6$ to IP$_5$ ratio was found in the soil with the highest SOM content,
suggesting a possible stabilization of IP$_6$ due to association with SOM (Makarov et al., 1997; Borie et al., 1989).
In contrast, the Ferralsol sample containing high amounts of Fe and Al showed the smallest IP$_6$ to IP$_5$ ratio, even
though IP$_6$ is known to strongly adsorb to sesquioxides (Anderson et al., 1974; Anderson and Arlidge, 1962).
However, the production, input and mineralisation rates of IP$_6$ and IP$_5$ are not known for our soil samples. Further
research is needed to understand the mechanisms controlling the flux of lower-order IP in soil.





In the Ferralsol and the Cambisol, there was an overall decrease in the concentration of $IP_6$ and $IP_5$ following
hypobromite oxidation compared to the untreated extracts. Since the main cause of resistance of IP to hypobromite
oxidation is that of steric hindrance, which generally decreases with decreasing phosphorylation state and
conformation of the phosphate groups (axial vs. equatorial), we assume that low recoveries of added *myo*-$IP_6$ is
due to losses of precipitated $P_{org}$ compounds during the precipitation and dissolution steps. This is supported by
the decrease in the concentration of orthophosphate following hypobromite oxidation compared to untreated
extracts. Therefore, quantities of IP as reported in the current study should be considered as conservative.
**4.3      Structural composition of phosphomonoesters in hypobromite oxidised soil extracts**
The NMR spectra on hypobromite oxidised soil extracts revealed the presence of sharp and broad signals in the
phosphomonoester region. Transverse relaxation experiments revealed a rapid decay of the broad signal compared
to the sharp peaks of $IP_6$, which support the hypothesis that the compounds causing the broad signal arise from P
compounds other than IP. These findings are consistent with that of McLaren et al. (2019), who probed the
structural composition of phosphomonoesters in untreated soil extracts. Overall, measured $T_2$ times in the current
study on hypobromite oxidised extracts were markedly longer compared to that on untreated extracts reported in
McLaren et al. (2019). This could be due to removal of other organic compounds by hypobromite oxidation in the
matrix and therefore a decrease in the viscosity of the sample. This would result in an overall  faster tumbling of
the molecules and hence an increased $T_2$ relaxation time. As reported by McLaren et al. (2019), calculations of the
broad signal's linewidth based on the $T_2$ times were considerably lower compared to that of the standard
deconvolution fitting (SDF). When applying the same calculations to our samples, the linewidth of the broad signal
at half height is on average 5.2 Hz based on the $T_2$ times. In contrast, the linewidths acquired from the SDF average
to 256.1 Hz. McLaren et al. (2019) suggested that the broad signal is itself comprised of more than one compound.
Our results are consistent with this view and therefore it is likely that the main cause of the broad signal is a
diversity of P molecules of differing chemical environments within this region, rather than the slow tumbling of
just one macromolecule.
Since a portion of the broad signal is resistant to hypobromite oxidation, this suggests the organic P is complex
and in the form of polymeric structures. The chemical resistance of the broad signal to hypobromite oxidation may
also indicate a high stability in soil (Jarosch et al., 2015). Annaheim et al. (2015) found that concentrations of the
broad signal remained unchanged between three different organic fertiliser strategies after 62 years of cropping.
In contrast, the organic P compounds annually added with the fertilisers were completely transformed or lost in
the slightly acidic topsoil of the field trial. The large proportion of the broad signal in the total organic P pool
demonstrates its importance in the soil P cycle.
Unexpectedly, the transverse relaxation times of orthophosphate were shorter than that of the broad signal. This
was similarly the case in an untreated NaOH-EDTA extract of a forest soil with the same origin as the Cambisol
as reported in McLaren et al. (2019). The authors hypothesised that this might be due to the sample matrix (i.e.
high concentration of metals and organic matter). Whilst these factors are likely to affect $T_2$ times, they do not
appear to be the main cause as the hypobromite oxidised extracts in the current study contained low concentrations
of organic matter and metals as a consequence of the isolation procedure. The fast decay of orthophosphate was
found across all four soil extracts with a diverse array of organic P concentrations and compositions of organic P
in the phosphomonoester region. Therefore, another possible explanation could be a matrix effect or an association
with large organic P compounds causing the broad signal (McLaren et al., 2019). It is known that dynamic



intramolecular processes as ring inversion and intermolecular processes such as binding of small-molecule ligands
to macromolecules can cause a broadening or a doubling of resonances (Claridge, 2016). When the smaller
molecule is bound to the larger molecule, it experiences slower tumbling in the solution and hence a shorter $T_2$
time. It is possible that a chemical exchange of the orthophosphate with a compound in the matrix or an organic P
molecule could result in the short $T_2$ time of the orthophosphate peak. We carried out a $T_2$ experiment on a pure
solution of monopotassium phosphate (described in the Supporting Information), in which the matrix effects
should be considerably reduced compared to the soil extracts. We found that the $T_2$ time of orthophosphate
(203 ms) in the pure solution was considerably longer than that reported in soil extracts following hypobromite
oxidation.
**5     Conclusion**
Inositol phosphates are an important pool of organic P in soil, but information on the mechanisms controlling their
flux in soil remain limited due in part to an inability to detect them using solution [31]P NMR spectroscopy. For the
first time, we identified six different lower-order IP in the solution [31]P NMR spectra on soil extracts. Solution [31]P
NMR spectra on hypobromite oxidised extracts revealed the presence of up to 70 sharp peaks, which about 50 %
could be identified. Our results indicate that the majority of the sharp peaks in solution [31]P NMR soil spectra were
resistant to hypobromite oxidation, and therefore suggest the presence of diverse IP. Our study highlights the
abundance of IP in soils and therefore their importance in terrestrial P cycles. Furthermore, we provide new insight
on the large pool of phosphomonoesters represented by the broad signal, of which a considerably portion was
resistant to hypobromite oxidation. Further research is needed to understand the chemical composition of the broad
signal, and the mechanisms controlling its flux in terrestrial ecosystems.
**Data availability**
All data presented in this study and the Supplement is also available by request from the corresponding author.
**Author contribution**
The experimental design was planned by JR, TM, DZ, RV and EF. The experiments were carried out by JR under
supervision of TM, DZ and RV. RV provided the MATLAB code for the standard deconvolution fitting of the
NMR spectra. The data was processed, analysed and interpreted by JR with support from TM, DZ and RV. JR
prepared the manuscript with contributions from all co-authors.
**Conflicts of interest**
The authors declare that they have no conflict of interest.
**Acknowledgements**
The authors are grateful to Dr Laurie Paule Schönholzer, Dr Federica Tamburini, Mr Björn Studer, Ms Monika
Macsai, and Dr Charles Brearley for technical support. Furthermore, the authors thank Dr Astrid Oberson, Dr



David Lester, Dr Chiara Pistocchi and Dr Gregor Meyer for providing soil samples. This study would not have
been possible without the IP standards originating from the late Dr Dennis Cosgrove collection and Dr Max Tate
collection, which we highly appreciate. We gratefully acknowledge funding from the Swiss National Science
Foundation [grant number 200021_169256].
**Financial support**
This project was funded by the Swiss National Science Foundation, Grant 200021_169256.



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



**Table 1. General characteristics of soil samples used in this study.**

| Soil type | - | Ferralsol | Vertisol | Cambisol | Gleysol |
|---|---|---|---|---|---|
| Country | - | Colombia | Australia | Germany | Switzerland |
| Coordinates sampling site | - | 4°30' N / 71°19' W | 27°52' S / 151°37' E | 50°21' N / 9°55' E | 47°05' N / 8°06' E |
| Elevation | m ASL | 150 | 402 | 800 | 612 |
| Sampling depth | cm | 0-20 | 0-15 | 0-7 | 0-10 |
| Year of sampling | year | 1997 | 2017 | 2014 | 2017 |
| Land use | - | Pasture | Arable field | Forest | Pasture |
| $C_{tot}$ | g C/kg$_{soil}$ | 26.7 | 23.9 | 90.3 | 148.3 |
| $N_{tot}$ | g N/kg$_{soil}$ | 1.7 | 1.9 | 6.6 | 10.9 |
| pH in $H_2O$ | - | 3.6 | 6.1 | 3.6 | 5.0 |




**Table 2. Standard solutions used for the spiking experiment of the hypobromite oxidised soil extracts. All standards**
**were dissolved in 0.25 M NaOH and 0.05 M $Na_2EDTA$.**

| Standard | Product number | Company/origin | Concentration of standard in NaOH-EDTA (mg/mL) |
|---|---|---|---|
| *myo*-$IP_6$ | P5681 | Merck (Sigma-Aldrich) | 8.10 |
| L-*chiro*-$IP_6$ | Collection of Dr Max Tate | | 2.39 |
| D-*chiro*-$IP_6$ | CAY-9002341 | Cayman Chemical | 2.00 |
| *neo*-$IP_6$ | Collection of Dr Dennis Cosgrove, made up in 15 mM HCl | | 4.62 |
| D-*myo*-(1,2,4,5,6)-$IP_5$ | CAY-10008452-1 | Cayman Chemical | 2.00 |
| *myo*-(1,2,3,4,6)-$IP_5$ | 93987 | Merck (Sigma-Aldrich) | 2.00 |
| D-*myo*-(1,3,4,5,6)-$IP_5$ | CAY-10009851-1 | Cayman Chemical | 2.00 |
| D-*myo*-(1,2,3,5,6)-$IP_5$ | CAY-10008453-1 | Cayman Chemical | 2.00 |
| *scyllo*-$IP_5$ | Collection of Dr Dennis Cosgrove | | 2.64 |
| L-*chiro*-$IP_5$ | Collection of Dr Dennis Cosgrove | | 2.24 |
| *neo*-$IP_5$ | Collection of Dr Dennis Cosgrove | | 2.45 |
| *myo*-$IP_4$ | Collection of Dr Dennis Cosgrove | | 2.76 |
| *scyllo*-$IP_4$ | Collection of Dr Dennis Cosgrove | | 2.41 |
| *neo*-$IP_4$ | Collection of Dr Dennis Cosgrove | | 2.33 |






**Table 3. Concentrations of total P as measured by XRF and 0.25 M NaOH + 0.05 M EDTA extractable P before and**
**after hypobromite oxidation of soil extracts. Concentrations of total P in NaOH-EDTA extracts were determined by**
**ICP-OES, whereas that of molybdate reactive P (MRP) was determined by the malachite green method of Ohno and**
**Zibilske (1991). Concentrations of molybdate unreactive P (MUP) were calculated as the difference between total P and**
**MRP.**

| Measure | | Ferralsol | Vertisol | Cambisol | Gleysol |
|---|---|---|---|---|---|
| **XRF** | $P_{tot}$ (mg P/kg$_{soil}$) | 320 | 1726 | 3841 | 2913 |
| **NaOH-EDTA extractable P (untreated)** | $P_{tot}$ (mg P/kg$_{soil}$) | 160 | 484 | 1850 | 1490 |
| | MRP (mg P/kg$_{soil}$) | 67 | 351 | 525 | 610 |
| | MUP ($P_{org}$) (mg P/kg$_{soil}$) | 93 | 133 | 1326 | 880 |
| **NaOH-EDTA extractable P (hypobromite oxidised)** | $P_{tot}$ (mg P/kg$_{soil}$) | 77 | 158 | 580 | 578 |
| | MRP (mg P/kg$_{soil}$) | 32 | 111 | 283 | 231 |
| | MUP ($P_{org}$) (mg P/kg$_{soil}$) | 45 | 47 | 297 | 348 |






**Table 4. Concentrations (mg P/kg$_{soil}$) of P compounds in solution $^{31}$P NMR spectra of 0.25 M NaOH + 0.05 M EDTA**
**soil extracts (Ferralsol, Vertisol, Cambisol and Gleysol) before and after hypobromite oxidation (HO). Quantification**
**was based on spectral integration and deconvolution fitting. The proportion of P detected in hypobromite oxidised**
**extracts compared to that in untreated extracts is provided in brackets.**

| Phosphorus class | | Ferralsol | Vertisol | Cambisol | Gleysol |
|---|---|---|---|---|---|
| **Phosphonates** | before HO | 1.0 | 2.6 | 14.5 | - |
| | after HO | - | - | 3.0 (21) | 0.2 |
| **Orthophosphate** | before HO | 54.8 | 221.4 | 434.3 | 368.3 |
| | after HO | 32.0 (58) | 116.6 (53) | 329.3 (76) | 243.4 (66) |
| **Phosphomonoester** | before HO | 36.3 | 39.1 | 501.1 | 399.2 |
| | after HO | 12.7 (35) | 24.2 (62) | 210.3 (42) | 292.1 (73) |
| **Broad peak** | before HO | 21.6 | 30.9 | 305.8 | 216.7 |
| | after HO | 8.3 (39) | 19.3 (63) | 99.2 (32) | 108.4 (50) |
| **Phosphodiester** | before HO | 5.1 | - | 28.2 | 26.9 |
| | after HO | - | - | - | 2.0 (8) |
| **Pyrophosphate** | before HO | 1.9 | 1.8 | 12.9 | 23.9 |
| | after HO | - | - | - | - |




**Table 5. Concentrations of identified inositol phosphates (IP) in hypobromite oxidised 0.25 M NaOH + 0.05 M EDTA**
**soil extracts (Ferralsol, Vertisol, Cambisol and Gleysol). Concentrations were calculated from solution $^{31}$P NMR spectra**
**using spectral deconvolution fitting including an underlying broad signal. When no concentration is given, the IP**
**compound was not detected in the respective soil extract. Chemical shift positions are based on the NMR spectrum of**
**the Cambisol extract (Fig. SI8 in the Supporting Information). Peak positions varied up to +0.018 ppm (Gleysol).**
**Conformation equatorial (eq) and axial (ax) according to Turner et al. (2012).**

| Phosphorus compound | Chemical shift δ ppm | Concentrations (mg P/kg_soil) | | | |
|---|---|---|---|---|---|
| | | Ferralsol | Vertisol | Cambisol | Gleysol |
| *myo*-IP$_6$ | 4.97, 4.06, 3.70, 3.57 | 1.1 | 0.6 | 26.3 | 85.0 |
| *scyllo*-IP$_6$ | 3.20 | 0.4 | 0.3 | 15.6 | 41.1 |
| *neo*-IP$_6$ 4-eq/2-ax | 5.86, 3.75 | - | - | 1.4 | 8.8 |
| *neo*-IP$_6$ 2-eq/4-ax | 4.36, 4.11 | - | - | 4.0 | 1.3 |
| *chiro*-IP$_6$ 2-eq/4-ax | 5.66, 4.25, 3.83 | - | - | 9.4 | 8.6 |
| *myo*-(1,2,4,5,6)-IP$_5$ | 4.42, 3.97, 3.72, 3.36, 3.25 | - | - | 7.0 | 4.1 |
| *myo*-(1,3,4,5,6)-IP$_5$ | 4.12, 3.60, 3.23 | - | - | 2.8 | 1.3 |
| *scyllo*-IP$_5$ | 3.81, 3.31, 3.05 | 0.7 | 0.5 | 10.8 | 6.1 |
| *neo*-IP$_5$ | 4.64, 4.27, 4.01, 3.87, 3.13 | - | - | 3.3 | 2.1 |
| *chiro*-IP$_5$ | 4.61, 3.39 | - | - | 0.9 | - |
| *scyllo*-(1,2,3,4)-IP$_4$ | 4.12, 3.25 | 0.8 | - | 4.3 | 1.0 |
| **Total IP** | | 3.0 | 1.4 | 85.9 | 159.3 |






**Table 6. Transversal relaxation times (T₂) of various P species in the orthophosphate and phosphomonoester regions as**
**determined by solution ³¹P nuclear magnetic resonance (NMR) spectroscopy and a Carr-Purcell- Meiboom-Gill**
**(CPMG) pulse sequence on hypobromite oxidised soil extracts.**

| Phosphorus compound | $T_2$ [ms] | | | |
|---|---|---|---|---|
| | **Ferralsol** | **Vertisol** | **Cambisol** | **Gleysol** |
| *myo*-IP$_6$ | 163 | 140 | 139 | 121 |
| *scyllo*-IP$_6$ | 250 | 155 | 154 | 144 |
| *neo*-IP$_6$ | - | - | 203 | 102 |
| *D-chiro*-IP$_6$ | - | - | 108 | 132 |
| **orthophosphate** | 14 | 9 | 17 | 6 |
| **broad peak** | 44 | 69 | 89 | 62 |