# Peer review of "Identification of lower-order inositol phosphates (IP5 and IP4)"

_Biogeosciences, 2019_

## Referee Comment (RC1) · Anonymous Referee #1 · 25 Jan 2020

I am happy to see this study on the inositol phosphate stereoisomers in soils, particularly the lower-order esters. The inositol phosphates are a quantitatively important and ecologically interesting group of phosphorus compounds in soils, but much remains unknown. This study uses hypobromite oxidation and solution 31P NMR spectroscopy to identify inositol phosphate stereoisomers in four soils. The spectroscopic work is of high quality. The presence of the higher-order stereoisomers is well-established, but this work identifies several lower-order esters in various stereoisomeric forms. Although these have been reported previously by chromatography, and inferred in NMR

studies based on resistance to bromination, this is the first direct identification by solution 31P NMR. I recommend publication, but ask the authors to consider the following comments in their revision.

1. Hypobromite oxidation destroys organic matter except the inositol phosphates, but this statement seems true only for the higher-order esters. The hexaphosphates definitely resist bromination (e.g. Turner et al. 2012). However, it seems that earlier papers on the method suggested at least partial decomposition of the pentakisphosphates and complete decomposition of other esters. If these compounds persisted here, particularly the tetrakisphosphates, this suggests the possibility that oxidation was incomplete (see below). Did the authors test the resistance of the target compounds to bromination? If not, it might be worth adding a statement about the extent to which the lower esters are expected to resist bromination.

2. There appears to be a couple of problems with the bromination procedure here. First, it appears that there was incomplete oxidation, with persistence of some diesters, phosphonates, inositol tetrakisphosphates, and the broad signal (assuming it represents high molecular weight organic matter). Second, and as discussed by the authors, there appears to have been considerable loss of phosphorus during bromination, perhaps through precipitation, as indicated by a loss of orthophosphate, pyrophosphate, and the inositol hexakisphosphates. Inorganic phosphate should increase markedly following bromination, as organic phosphates are destroyed and converted to inorganic orthophosphate. This isn't a problem for identification, but represents a problem for the quantification of compounds in the brominated extracts, at least if these values are to represent concentrations of the identified forms in the original soil. Given the precipitation issue, the concentrations in brominated extracts should probably be considered unreliable, and it'd be better to give quantitative values only from those signals identified in the unbrominated extracts. Data from the brominated extracts are of course still useful as qualitative identifications.

3. It has been claimed that inositol phosphates account for a negligible amount of soil

organic phosphorus and that their importance in the soil has been over-emphasized in the literature. This argument was made sufficiently strongly by one group that a prominent mycorrhizal ecologist, now sadly deceased, rewrote the section on inositol phosphate utilization by ectomycorrhizal fungi in her influential textbook. The authors might consider mentioning this in the discussion section, given the relatively large concentrations of inositol phosphates they detected in their soils.

4. The 'broad signal' is supposed to consist of high molecular weight organic compounds. These should be destroyed by hypobromite oxidation. If not, this suggests that either (1) the oxidation was incomplete, or (2) the broad signal is caused by something else other than high molecular weight compounds. The authors might comment on this.

5. Related to the broad signal, I think it would be worth explaining a little more about the deconvolution procedure used here. Some recent studies appear to have deconvoluted from the baseline to the top of the peaks in the monoester region, which is certain to overestimate the proportion of each signal. This might in turn exaggerate differences between signals in brominated unbrominated extracts, given that the 'broad signal' appears to be reduced by bromination.

Line-by-line comments:

Line 12: most studies have identified inositol phosphates by NMR in recent decades, not chromatography. Perhaps you refer specifically to lower esters, in which case perhaps state this at the start of the sentence.

Line 17: shouldn't the 'broad signal' be destroyed by hypobromite oxidation?

Line 20: I understood that one of the myo-IP5 forms (myo-inositol-1,3,4,5,6) is supposed to be rare in nature and therefore unlikely to occur in soils. This is because phytases cleave phosphates other than the C-2 phosphate, often leaving myo-inositol-2-phosphate as the final product. It's therefore a surprise to see this compound detected in two of the soils here. Could the authors comment on this?

Line 43: this is only partially correct – pigs are monogastrics, but phytate is still hydrolyzed during passage through the animal – probably in the hindgut – so pig manure tends to contain little phytate. See for example: Leytem, A. B., B. L. Turner, and P. A. Thacker. 2004. Phosphorus composition of manure from swine fed low-phytate grains: Evidence for hydrolysis in the animal. Journal of Environmental Quality 33:2380-2383. Turner, B. L., and A. B. Leytem. 2004. Phosphorus compounds in sequential extracts of animal manures: chemical speciation and a novel fractionation procedure. Environmental Science and Technology 38:6101-6108.

Line 76: perhaps add 'and a chelating agent' – the EDTA is important in the single-step extraction.

Line 80: this was presumably the case in Turner and Richardson 2004, who presented chemical shifts of lower scyllo-IP esters, but did not detect the corresponding signals in NMR spectra of soil extracts.

Line 97: it's not clear why these four soils were chosen for study – perhaps add a brief explanation.

Line 118: This sentence seems redundant if the method was the same. Delete?

Line 121: Turner recently published the hypobromite method as a chapter in the new book on inositol phosphate methods, which might be appropriate to cite here: Turner, B. L. 2020. Isolation of inositol hexakisphosphate from soils by alkaline extraction and hypobromite oxidation. Pages 39-46 in G. J. Miller, ed. Inositol Phosphates: Methods and Protocols. Springer US, New York, NY.

Line 190 and 221: Please provide more information on the deconvolution procedure. Some recent studies appear to have deconvoluted from the baseline to the top of the peaks in the monoester region, which is certain to overestimate the proportion of each signal. This might in turn lead to differences between signals in brominated unbromi-

nated extracts.

Line 262: What could the broad signal possibly be, in brominated extracts?

Line 225: comma instead of period. The persistence of some phosphodiesters suggests incomplete oxidation.

Line 276: this depends on how spectra were deconvoluted – see point above.

Line 278: It's interesting to see evidence for the two conformers of neo-IP6. The proportion of the two conformers is definitely related to pH – is it possible that pH was <12 in the extracts, promoting the presence of the two forms?

Line 283: Aren't lower-order esters destroyed by bromination?

Line 292: Turner and Richardson 2004 reported signals for two different scyllo-IP4 compounds. Signals from these were not identified in brominated soil extracts, but resolution was not as high as in this study. It looks like only a single scyllo-IP4 isomer was assessed here, so perhaps scyllo-IP4 is underestimated (assuming that the other scyllo-IP4 isomer occurs in soils, and that the tetraakisphosphates resist bromination).

Line 311: 6 in subscript.

Line 327: orthophosphate should increase following bromination, as organic phosphates are converted to inorganic orthophosphate. This indicates precipitation or loss of phosphates in some other way during the bromination procedure.

Line 404: also along the Haast chronosequence: Turner, B. L., A. Wells, and L. M. Condron. 2014. Soil organic phosphorus transformations along a coastal dune chronosequence under New Zealand temperate rain forest. Biogeochemistry 121:595-611. The Baker study on the Franz Josef involved the same sites as Turner et al. 2007, so the separate statement on the Baker study could probably be deleted and the citation rolled into with the others.

Line 418: see above. I think the concentrations on the brominated extracts should be

considered unreliable, given the apparent loss of phosphorus during the procedure. It'd probably be better to focus on quantitative values from comparable signals in the unbrominated extracts, and give information from the brominated extracts as qualitative identifications.

Line 434: My impression is that the complexity of the monoester region means that deconvolution of all signals could easily account for the apparent broad signal. How does the possibility of more than one compound affect the accuracy of the deconvolution based on a single broad signal?

Line 436: This paragraph is awkward. First, the broad signal is supposedly made up of high molecular weight organic matter, which should be destroyed by bromination. Second, whether the compound forming the broad signal (or compounds, if they exist) occur in the soil is open to question – most scientists working on soil organic matter now accept that much of the high molecular weight material in alkaline soil extracts is formed as an artifact of the extraction procedure. Finally, the statement that the broad signal didn't change after 62 years of cropping seems to indicate precisely the opposite interpretation to that of the authors – that it demonstrates its importance in the soil P cycle. If it's so stable that it never changes, that suggests to me that it's actually fairly unimportant, at least ecologically or agronomically.

Table 3 – you could combine this table with Table 1 to streamline display items.

Table 4 – indicate that the broad peak also represents phosphomonoesters.

Table 5 – I think it's fairly safe to assume that the chiro-IP6 is the D form, given that L-chiro-inositol has never been detected in phosphorylated form in nature. Also it's interesting to see from this table that the neo+D-chiro-IP6 and the majority of the lower-order esters were detected only in two of the four soils. I didn't get this impression from reading the text.

Table S1 – this indicates a considerable proportion of the phosphorus has been lost

during the bromination procedure.

---

## Referee Comment (RC2) · Anonymous Referee #2 · 28 Feb 2020

The objective of this manuscript was to characterize and quantify inositol phosphates (IP) in soil extracts following hypobromite oxidation using 31P nuclear magnetic resonance (P NMR) spectroscopy. This is a very technical paper with respect to the chemical methods utilized. Given that the mandate of this journal is: "interactions between the biological, chemical, and physical processes in terrestrial or extraterrestrial life with the geosphere, hydrosphere, and atmosphere. The objective of the journal is to cut across the boundaries of established sciences and achieve an interdisciplinary view of these interactions" (from the journal website)", this paper does not seem like a good

fit for the journal. While the authors identified a wide range of different P compounds in their four soil samples, no attempt was made to relate these compounds back to broader biological, chemical or physical processes within these soils. As such, it will not be of interest to the majority of Biogeosciences readers, as currently written, and will likely be overlooked by the scientists who would be interested in such a technical paper. In my opinion, this would be a better fit in either an environmental chemistry journal or in the chemistry section of a soil science journal. Thus, in my opinion the authors should withdraw this paper from this journal and submit it to another journal that better fits the paper's focus. If the authors choose not to do this, then they must significantly revise the manuscript to keep it within the journal's scope, to clearly demonstrate the significance of these identified P compounds to P cycling in these soils, and to P cycling more broadly.

Abstract: As written, the abstract make it clear that this is chemistry methods paper, not a biogeochemical study, because the results and conclusions highlighted in the abstract indication only that the authors were able to identify these peaks, but make no reference to their relative importance in the studied soils and to P cycling in these and other soils. This supports my point above that this is not an appropriate journal for this paper as currently written. In addition the abstract needs to be more carefully edited, as it is awkwardly written in places. For example, lines 14-15: "include the A horizon of a Ferrasol from Columbia, of a Cambisol from Switzerland, of a Gleysol from Switzerland and of a Cambisol from Germany" should be "include A horizons from a Ferrosol (Columbia), a Cambisol and a Gleysol froom Switzerland, and a Cambisol from Germany". And why is the phrase "(using solution 31P NMR spectroscopy)" included in line 19, given that the method was given in line 13?

Introduction: The introduction provides a good overview of the chemical methodology for extracting and characterizing IP in soil, as would be expected for a chemical methods paper. It gives a very brief overview of the factors generally controlling IP in soils, but doesn't give much information about why there is a need to specifically characterize all of these different IP forms. What insights into soil P cycling would we gain from identifying these compounds that we don't already have by from the IP compounds we can already identify? And what information would be expected from analyzing them in different soils? And the hypothesis seems to be something that was tacked on at the end, and doesn't make a lot of sense: "We hypothesize that a large portion of sharp peaks in the phosphomonoester region of untreated soil extracts would be resistant to hypobromite oxidation, which would indicate the presence of IP". This again emphasizes that this is a chemical methods paper only. Other points in the Introduction: l. 35: "Riley Andrew et al., 2006)" why is the authors first name included (Andrew M. Riley is the first author of the paper)? This should be "Riley et al., 2006". And the listing in the References (l. 641-644) contains the first names of other authors of this paper. "Shears Stephen, B" should be "Shears, SB", and "Potter Barry VL" should be "Potter BVL". The correct names are very obvious when reading the manuscript, so I'm not sure why they are incorrect here. l. 39 and elsewhere in the text: when citing a list of references, it is conventional to list them in order from oldest to most recent. l. 87: "was resistant" should be "were resistant", because it modified "signals", which is plural.

Methods: As written, there is far too much technical information (e.g. about the transverse relaxation experiments), which will not be of any interest to the majority of readers of this journal. And other important information seems to be missing. See specific points listed below. Also, I believe that Turner has published a new paper of the hypobromite oxidation method. How does the method used compare to that method. l. 117: Please provide information on the total volume of extractant used and the total volume of filtrate produced, to help the reader put the hypobromite oxidation experiments into context. In line 121, it indicates that "10 mL of the filtrate was used". What proportion of the total filtrate is this – 10% or 100%? l. 144-145: This sentence is awkwardly written. Change "...in solution is that of molybdate unreactive P (MUP), which is considered to be largely that of organic P" to "in solution is molybdate unreactive P (MUP), which is predominantly organic P for these samples" l. 146-147: "a duplicate sample of the Cambisol and the Gleysol was spiked" should be "duplicate samples of the Cambisol

and Gleysol were spiked" l. 161-162: The inclusion of the Vestergren et al. 2012 paper here confused me. This group left their samples to sit overnight because they used a sulfide treatment to remove paramagnetic ions. Was this also done for the current study? If so, then please describe the sulfide treatment more clearly. If not, then it would be better to replace this reference with one that is more appropriate. l. 193-195: Something seems to be missing here for the measurement of N observability. Using PtotICP-OES only makes sense if the entire sample after freeze-drying was used for the NMR analysis. However, that does not seem to be the case for this study. While it appears that the total mass of lyophilized material was used for the brominated samples (l. 167-168), a set mass (120 g) of the non-brominated lyophilized material was used, with no indication of how much of the total lyophilized material this represents. The proportion of total mass used must be factored into the equation to correctly determine NMR observability. This would also explain the differences in observability reported in the supplementary information (SI) for the brominated and unbrominated samples. l. 206-225: There is no need to include this much detail about the transverse relaxation papers. As noted above, the majority of readers of this paper in this journal will not be interested in these details. In addition, this appears to be a repeat of what was done for the McLaren et al. 2019 study. As such, all that is needed is to cite the previous publication. If the authors really thing this much detail is needed, it could be included in the SI. L. 226-233: Why are methods for statistical analyses reported here, when no results of statistical analysis are included in the Results, Discussion or SI?

Results: 1. Please provide spectra showing the entire spectrum for each brominated and unbrominated sample, scaled to allow the reader to see the full height of orthophosphate and the relative heights of other peaks compared to orthophosphate. All of the spectra currently in the manuscript show the monoester region only, with the orthophosphate peak truncated. This is needed to get a full sense of all the peaks for each sample, especially for the brominated samples. 2. The usefulness of the spectra shown in Fig. 3 are not clear. I am used to looking at NMR spectra, and I found these confusing, as with the exception of the Gleysol the red lines show little but noise. Again,

this would be appropriate for a chemical methods paper, emphasizing that this is not the best journal for this study. 3. I am concerned that the authors report signals for non-IP compounds in their brominated spectra. In my experience with this technique, if there are any peaks for non-IP compounds, that suggests that the oxidation was incomplete. And that in turn raises questions about the authors' assignment of peaks in the brominated samples. 4. How confident are the authors that all of the peaks were present in their soils prior to extraction and hypobromite oxidation? Isn't it possible that bromination degraded some high IPs (e.g. IP6) to lower IPs (IP5 and IP4)? The recovery of the added myo-IP6 was only 20 and 47%, which suggests it may have been degraded. l. 255: change "Although," to "However," l. 273: "A detailed view of the phosphomonoester region of spiked extracts is shown" should be "Detailed views of the phosphomonoester regions of spiked samples are shown" l. 306-316: I do not see the need to include any of this information about spin-echo analysis of selected P compounds in the current paper, as it will not be of any interest to the majority of readers of this paper in this journal.

Discussion: The P-NMR literature cited in this section seems biased to papers by the Smernik group. I have concerns about this because that group prepared their samples for NMR differently from most other groups, and from what was done for the current study. As such, results from that group may not be directly comparable here. In addition, it shows an unfamiliarity with the broader P-NMR literature, which is of concern. In general, however, I think the authors have done a reasonable job of trying to relate these P compounds to the literature and to the soils, which would be suitable to this journal. However, they should note the overall small proportion of total P that some of these compounds comprise. Are compounds in such low concentrations really an integral component of P cycling. And in my opinion, section 4.3 is not appropriate for this journal and would not be of interest to the majority of readers, and so should be cut. l. 322-324: Other studies have looked at what was not extracted by NaOH-EDTA, including with acid extraction after NaOH-EDTA or with solid-state P-NMR. See for example studies by He et al. These would be more appropriate to cite here than

McLaren et al., 2015a l. 333-334: "This will result in the production of carbon dioxide and simple organic acids" This sentence does not seem to be relevant here. How is this related to P? l. 340-342: If the authors had not shown peaks other than monoesters and orthophosphate, I might agree with them that the peaks in the monoester region are all IP. However, it is clear from the results they have shown that they did not have complete oxidation of all P compounds. So how can they be confident that they only have IP in the monoester region? This must be addressed. l. 348-350: I'm confused by the some of the papers cited here. Why are studies that did not use chromatography cited here to make a point about chromatography. Please rephrase, or remove the non-chromatography references. l. 356-363: As noted above, the authors did not have compete oxidation of all non-IP compounds in their extracts. So how can they be certain that this peak at 4.36 is an IP compound and not $\alpha$-glycerol. In addition, other groups have reported a peak that sits very close to $\alpha$-glycerol, and have urged caution about identifying this peak without spiking. This emphasizes a need for a broader review of the literature than just papers from the Smernik group.

l. 370: change "extracts, which the" to "extracts, of which the" l. 383: add spaces between the numbers and words here: "1axial" should be "1 axial" or "1-axial", etc.

---

## Author Comment (AC1) · 4 May 2020

**REVIEWER REPORT 1**

**Comment 1**
I am happy to see this study on the inositol phosphate stereoisomers in soils, particularly the lower-order esters. The inositol phosphates are a quantitatively important and ecologically interesting group of phosphorus compounds in soils, but much remains unknown. This study uses hypobromite oxidation and solution 31P NMR spectroscopy to identify inositol phosphate stereoisomers in four soils. The spectroscopic work is of high quality. The presence of the higher-order stereoisomers is well-established, but this work identifies several lower-order esters in various stereoisomeric forms. Although these have been reported previously by chromatography, and inferred in NMR studies based on resistance to bromination, this is the first direct identification by solution 31P NMR. I recommend publication, but ask the authors to consider the following comments in their revision.

**Response 1**
We thank the reviewer for the positive comments.

**Comment 2**
Hypobromite oxidation destroys organic matter except the inositol phosphates, but this statement seems true only for the higher-order esters. The hexaphosphates definitely resist bromination (e.g.Turner et al. (2012)). However, it seems that earlier papers on the method suggested at least partial decomposition of the pentakisphosphates and complete decomposition of other esters. If these compounds persisted here, particularly the tetrakisphosphates, this suggests the possibility that oxidation was incomplete (see below). Did the authors test the resistance of the target compounds to bromination? If not, it might be worth adding a statement about the extent to which the lower esters are expected to resist bromination.

**Response 2**
The main reaction pathway of the hypobromite oxidation procedure is the oxidation of organic matter and not its bromination. Our study is based on existing publications using hypobromite oxidation to isolate IPs. However, the action of hypobromite oxidation on each IP species, and also on 'organic matter', has not been clearly determined. The resistance of IP to hypobromite oxidation is considered to be due to increased steric hindrance and the high charge density of the organic molecule. Hence, the resistance of lower order IP to hypobromite oxidation decreases with decreasing number of phosphate groups bound to the molecule. We agree with the reviewer that it is possible some $IP_4$ was partially oxidised to $IP_{3-1}$. We have made this clearer in the body text.

We inserted the sentence (Lines 298-301): This could possibly be due to the partial dephosphorylation of *myo*-$IP_4$ during the hypobromite oxidation procedure. The reason of the reduced resistance of lower-order IP to hypobromite oxidation compared to $IP_{5+6}$ might be due to their reduced steric hindrance and charge density, as less phosphate groups are bound to the inositol ring.

Lastly, we note that Irving and Cosgrove (1981) reported inositol hexa- and penta-kisphosphates were resistant to hypobromite oxidation. Furthermore, in the current study, several peaks assigned to hexa- and pentakisphosphates in the hypobromite oxidised extracts were also present in the untreated extracts. Whilst the absolute concentration of these IPs may be questioned, we provide supporting evidence for their presence, which can be easily identified using solution $^{31}P$ NMR spectroscopy.

**Comment 3**
There appears to be a couple of problems with the bromination procedure here. First, it appears that there was incomplete oxidation, with persistence of some diesters,

phosphonates, inositol tetrakisphosphates, and the broad signal (assuming it represents high molecular weight organic matter). Second, and as discussed by the authors, there appears to have been considerable loss of phosphorus during bromination, perhaps through precipitation, as indicated by a loss of orthophosphate, pyrophosphate, and the inositol hexakisphosphates. Inorganic phosphate should increase markedly following bromination, as organic phosphates are destroyed and converted to inorganic orthophosphate. This isn't a problem for identification, but represents a problem for the quantification of compounds in the brominated extracts, at least if these values are to represent concentrations of the identified forms in the original soil. Given the precipitation issue, the concentrations in brominated extracts should probably be considered unreliable, and it'd be better to give quantitative values only from those signals identified in the unbrominated extracts. Data from the brominated extracts are of course still useful as qualitative identifications.

**Response 3**
The ratio of soil extract to bromine used in previous studies were 50 (Turner et al., 2012), 25 (Turner and Richardson, 2004), 20 to 10 (Turner, 2020), and 10 (Almeida et al., 2018). Consequently, the ratio of volume of soil extract to bromine used in the current study (16.7) is similar and at the higher end of that reported in previous studies. Nevertheless, we carried out a pilot study to test different soil extract to bromine ratios on spectral quality in the Gleysol soil, which had the highest organic matter content among the soils analysed in the current study: ratios covered 50.0, 25.0, 16.7, and 12.5. Solution $^{31}$P NMR spectroscopy on the hypobromite oxidised soil extracts revealed the overall peak diversity and intensity was highest for the 16.7 ratio (i.e. 0.6 mL $Br_2$ addition) (see Figure 1). Furthermore, we added a *myo*-$IP_6$ standard of known concentration to the Gleysol extract prior to hypobromite oxidation at the aforementioned ratios. These results showed that the recovery of added *myo*-$IP_6$ was highest (38%) for the 16.7 ratio compared to the 25.0 ratio (31%) or 12.5 ratio (32%). Of course, a problem with continuing to decrease the ratio of soil extract to bromine is that further oxidation of IP may occurs.

Unfortunately, previous studies have not reported quality assurance/control data for the ratio of soil extract to bromine. Nevertheless, solution $^{31}$P NMR spectra on hypobromite oxidised extracts in previous studies appear to show a broad signal in the phosphomonoester region based on a visual assessment: see Figure 3 in Turner et al. (2012) and Figure 3 in Turner and Richardson (2004). The authors did not include an underlying broad signal in their spectral deconvolution process. However, the study of Reusser et al. (2020a) showed that the inclusion of a broad signal in the phosphomonoester region is important for accurate quantification of the overlying sharp signals (i.e. *myo*-$IP_6$).

The persistence of on average half the organic P compounds as part of the broad signal in the phosphomonoester region highlights their chemical stability. Please also see Response 8 for more information.

The majority of NMR signals in the phosphodiester and phosphonate regions were removed following hypobromite oxidation. The small presence of some phosphodiesters or phosphonates in the Cambisol or Gleysol soils was interesting, but their identity is unclear. It is possible that a portion of these compounds may be protected from oxidation due to their complexation with other organic molecules and metals.

We consider the quantification of lower order IP in soil extracts following hypobromite oxidation to be a conservative estimation. This was stated in the body text (lines 435-441). Whilst the reviewer is correct that some lower-order IP may have been oxidised, these extracts also have the advantage of reduced signal overlap, which facilitates peak assignment and spectral fitting.

[Figure]

**Figure 1.** Solution $^{31}$P nuclear magnetic resonance (NMR) spectra (500 MHz) of the orthophosphate and phosphomonoester region of hypobromite oxidised 0.25 M NaOH + 0.05 M EDTA Gleysol extract, using 0.2 mL, 0.4 mL, 0.6 mL and 0.8 mL $Br_2$ in the hypobromite oxidation procedure. Signal intensities were normalised to the MDP peak (intensity of 1 on y-axes).

We added Figure 1 to the Supporting Information (Figure SI9), referring to it in the body text: Line 131-133: The optimal volume of $Br_2$ for oxidation was assessed in a previous pilot study using 0.2, 0.4, 0.6 and 0.8 mL $Br_2$ volumes, and then observing differences in their NMR spectral features (Figure SI9).

**Comment 4**
It has been claimed that inositol phosphates account for a negligible amount of soil organic phosphorus and that their importance in the soil has been over-emphasized in the literature. This argument was made sufficiently strongly by one group that a prominent mycorrhizal ecologist, now sadly deceased, rewrote the section on inositol phosphate utilization by ectomycorrhizal fungi in her influential textbook. The authors might consider mentioning this in the discussion section, given the relatively large concentrations of inositol phosphates they detected in their soils.

**Response 4**
It depends on the soil, some soils contain a relatively high proportion of organic P as phytate, others not. We think that the reviewer refers to Smith et al. (2008). In this textbook, the study of Smernik and Dougherty (2007) was cited, who reported that phytate concentrations comprised less than 5% of total organic P in Australian soils. In our study, IP comprised between 1% and 18% of the total pool of organic P in European soils. On the point of IP utilisation by ectomycorrhizal fungi, we do not believe our study addresses this aspect as even low concentrations of phytate could be considered important depending on turnover, and we would therefore prefer not to comment.

**Comment 5**
The 'broad signal' is supposed to consist of high molecular weight organic compounds. These should be destroyed by hypobromite oxidation. If not, this suggests that either (1) the oxidation was incomplete, or (2) the broad signal is caused by something else other than high molecular weight compounds. The authors might comment on this.

**Response 5**
Correct, a portion of phosphomonoesters as part of the 'broad signal' has been found with apparent high molecular size (McLaren et al., 2015; McLaren et al., 2019) and appears to be associated with soil organic matter (McLaren et al 2020). However, as discussed in Response 2, factors increasing the resistance to hypobromite oxidation are steric hindrance and high charge density of an organic compound. Consequently, the action of hypobromite oxidation on phosphomonoesters exhibiting a broad NMR signal is unknown as their structure is undefined.

In general, sugars and ribonucleotides can most certainly be destroyed by hypobromite oxidation. However, there could be molecules with high molecular weight which would not be oxidised, e.g. highly resistant organic pesticides. To test this, one would need to carry out hypobromite oxidation on known compounds of high molecular weight present in soil and evaluate their resistance. Unfortunately, the composition of high molecular weight material in soil is not fully understood. We do not believe that hypobromite oxidation was incomplete based on details provided in Response 3, and that $Br_2$ was present in excess and soil extracts were kept at reflux after $Br_2$ addition. Furthermore, we note that based on a visual assessment a broad signal was also present in soil extracts following hypobromite oxidation in previous studies (Turner and Richardson, 2004; Turner et al., 2012).

Since a broad signal was observed in the NMR spectra on hypobromite oxidised extracts, we wanted to understand its structural composition. We carried out transverse relaxation ($T_2$) experiments in order to determine if the broad signal itself was comprised of (i) a series of neighbouring sharp peaks, which would likely arise from small molecules such as IP, or (ii) one (or a few) broad peak(s), which would likely arise from complex structures of 'higher' molecular weight (Bloembergen et al .1948). Our results support the latter, which suggest the remaining NMR signal as part of the broad signal is comprised of molecules with larger apparent molecular size than IP. Please also see Response 27.

In our study, we also propose a third option, namely that the complex structure of the compounds as part of the broad signal following hypobromite oxidation is due to their 'protection' via metals or configuration which enhances steric hindrance. Please also see Response 8.

**Comment 6**
Related to the broad signal, I think it would be worth explaining a little more about the deconvolution procedure used here. Some recent studies appear to have deconvoluted from the baseline to the top of the peaks in the monoester region, which is certain to overestimate the proportion of each signal. This might in turn exaggerate differences between signals in brominated unbrominated extracts, given that the 'broad signal' appears to be reduced by bromination.

**Response 6**
The reviewer is correct. Studies that carry out spectral deconvolution by fitting sharp peaks from the peak maxima to the baseline will likely overestimate the proportion of sharp peaks. This was demonstrated in a recent study, which found that fitting a broad signal was needed for accurate quantification of organic P compounds (e.g. *myo*-IP$_6$) (Reusser et al., 2020b). In the current study, spectral deconvolution fitting was carried out with an underlying broad signal in the phosphomonoester region, as described in Reusser et al. (2020b). Briefly, we carried out scripts containing a non-linear optimization algorithm in MATLAB® R2017a (The MathWorks, Inc.) and fitted visually identifiable peaks by constraining their line-widths at half height as well as the lower and upper boundary of the peak positions. The sharp signals of high intensity (e.g. orthophosphate) and the broad peak were fitted using a Lorentzian lineshape, whereas sharp signals of low intensity were fitted using a Gaussian lineshape. We have made this clearer in the body text.

Inserted (Line 195-203): Due to overlapping peaks in the orthophosphate and phosphomonoester region, spectral deconvolution fitting (SDF) was applied as described in Reusser et al. (2020b). In brief, the SDF procedure involved the fitting of an underlying broad signal, based on the approach of Bünemann et al. (2008) and McLaren et al. (2019). We carried out the SDF with a non-linear optimisation algorithm in MATLAB® R2017a (The MathWorks, Inc.) and fitted visually identifiable peaks by constraining their line-widths at half height as well as the lower and upper boundary of the peak positions along with an underlying broad signal in the phosphomonoester region. The sharp signals of high intensity (e.g. orthophosphate) and the broad peak were fitted using Lorentzian lineshapes, whereas sharp signals of low intensity were fitted using Gaussian lineshapes.

**Line-by-line comments**

**Comment 7 (Line 12)**
most studies have identified inositol phosphates by NMR in recent decades, not chromatography. Perhaps you refer specifically to lower esters, in which case perhaps state this at the start of the sentence.

**Response 7**
We made this clearer in the text:

Changed from (Lines 10-12): This is because their quantification typically requires a series of chemical extractions, including hypobromite oxidation to isolate inositol phosphates, followed by chromatographic separation.

Changed to (Lines 10-12): This is because the quantification of lower-order IP typically requires a series of chemical extractions, including hypobromite oxidation to isolate IP, followed by chromatographic separation.

**Comment 8 (Line 17)**
shouldn't the 'broad signal' be destroyed by hypobromite oxidation?

**Response 8**
Please see Response 5. In addition, IP are considered to resist hypobromite oxidation due to steric hindrance and high charge density. The structural configuration and exact chemical nature of the compounds causing the broad signal in the phosphomonoester region is not known. Studies have shown that these compounds are of complex structure, apparent high molecular weight and resistant to enzymatic hydrolysis (Jarosch et al., 2015; McLaren et al., 2015; McLaren et al., 2019). Hence, as the chemical structure is unknown, its resistance to hypobromite oxidation could not be evaluated in advance. Nevertheless, our study shows that on average half of the organic P as part of the broad signal was oxidised following hypobromite oxidation. The remaining broad signal which is resistant to hypobromite oxidation suggests complex structures of high chemical stability. This has been stated in the manuscript (lines 462-464).

**Comment 9 (Line 20)**
I understood that one of the myo-IP$_5$ forms (myo-inositol-1,3,4,5,6) is supposed to be rare in nature and therefore unlikely to occur in soils. This is because phytases cleave phosphates other than the C-2 phosphate, often leaving myo-inositol-2-phosphate as the final product. It's therefore a surprise to see this compound detected in two of the soils here. Could the authors comment on this?

**Response 9**
myo-(1,3,4,5,6)-IP$_5$ was reportedly measured as the thermal decomposition product of a phytate standard (Doolette and Smernik, 2018). It is possible that myo-IP$_6$ undergoes transformation via abiotic means to myo-(1,3,4,5,6)-IP$_5$, which could then be adsorbed by soil

constituents. Alternatively, $myo$-(1,3,4,5,6)-$IP_5$ could have been added biologically. For example, Stephens and Irvine (1990) report $myo$-(1,3,4,5,6)-$IP_5$ as an intermediate in the synthesis of $IP_6$ from $myo$-IP in the cellular slime mould *Dictyostelium*. In addition, Sun et al. (2017) report $myo$-(1,3,4,5,6)-$IP_5$ to occur as part of a possible minor pathway in the degradation of $myo$-$IP_6$ by *Aspergillus niger* phytase and acid phosphatase from potato. Later, Sun and Jaisi (2018) reported the presence of $myo$-(1,3,4,5,6)-$IP_5$ in different animal feeds and manures. We have revised the manuscript accordingly:
Lines 396-402: It is possible that an abiotic transformation of $myo$-$IP_6$ to $myo$-(1,3,4,5,6)-$IP_5$ occurs, which could then be adsorbed by soil constituents. Stephens and Irvine (1990) reported $myo$-(1,3,4,5,6)-$IP_5$ as an intermediate in the synthesis of $IP_6$ from $myo$-IP in the cellular slime mould *Dictyostelium*. Therefore, $myo$-(1,3,4,5,6)-$IP_5$ could have been biologically added to the soil. Furthermore, $myo$-(1,3,4,5,6)-$IP_5$ was present in different animal feeds and manures (Sun and Jaisi, 2018). Sun et al. (2017) reported $myo$-(1,3,4,5,6)-$IP_5$ and $myo$-(1,2,4,5,6)-$IP_5$ as intermediates in the minor, resp. major pathways of Aspergillus niger phytase and acid phosphatase (potato) phytate degradation.

**Comment 10 (Line 43)**
this is only partially correct – pigs are monogastrics, but phytate is still hydrolyzed during passage through the animal – probably in the hindgut – so pig manure tends to contain little phytate. See for example: Leytem, A. B., B. L. Turner, and P. A. Thacker. 2004. Phosphorus composition of manure from swine fed low-phytate grains: Evidence for hydrolysis in the animal. Journal of Environmental Quality 33:2380-2383. Turner, B. L., and A. B. Leytem. 2004. Phosphorus compounds in sequential extracts of animal manures: chemical speciation and a novel fractionation procedure. Environmental Science and Technology 38:6101-6108.

**Response 10**
We agree that the study of Leytem et al. (2004) indicates that phytate can be hydrolysed during passage through the animal. However, the authors did not measure lower order IP in their samples. Therefore, it is not known if a complete hydrolysis of phytate occurred or if $IP_6$ was hydrolysed to $IP_5$. We added this to the manuscript along with referring to transgenic pigs:

Changed from (lines 42-44): However, the addition of $myo$-$IP_6$ to soil can also occur via manure input because monogastric animals are incapable of digesting $myo$-$IP_6$ without the addition of phytases to their diets (Leytem and Maguire, 2007; Turner et al., 2007).

Changed to (lines 42-46): However, the addition of $myo$-$IP_6$ to soil can also occur via manure input because monogastric animals are mostly incapable of digesting $myo$-$IP_6$ without the addition of phytases to their diets (Leytem and Maguire, 2007; Turner et al., 2007). An exception to this are pigs, which were found to at least partially digest phytate (Leytem et al., 2004), and transgenic pigs expressing salivary phytase (Golovan et al., 2001; Zhang et al., 2018).

**Comment 11 (Line 76)**
perhaps add 'and a chelating agent' – the EDTA is important in the single-step extraction.

**Response 11**
Agreed, we added 'and a chelating agent'.

**Comment 12 (Line 80)**
this was presumably the case in Turner and Richardson 2004, who presented chemical shifts of lower scyllo-IP esters, but did not detect the corresponding signals in NMR spectra of soil extracts.

**Response 12**
The author's assessment of the study by Turner and Richardson (2004) may be correct, which is discussed using their more recent study (Turner et al., 2012) in the following section. Other possible reasons are a low signal-to-noise ratio of their NMR spectra using their experimental procedure, or a focus on $IP_6$ rather than lower-order IP. We would prefer not to speculate in the manuscript, and have not made any changes.

**Comment 13 (Line 97)**
it's not clear why these four soils were chosen for study – perhaps add a brief explanation.

**Response 13**
Agreed, we inserted the sentence (Lines 103-104): The four soil samples were chosen from a larger collection based on their diverse concentration of $P_{org}$ and composition of the phosphomonoester region in NMR spectra (Reusser et al., 2020b).

**Comment 14 (Line 118)**
This sentence seems redundant if the method was the same. Delete?

**Response 14**
Agreed, we have deleted the sentence.

**Comment 15 (Line 121)**
Turner recently published the hypobromite method as a chapter in the new book on inositol phosphate methods, which might be appropriate to cite here: Turner, B. L. 2020. Isolation of inositol hexakisphosphate from soils by alkaline extraction and hypobromite oxidation. Pages 39-46 in G. J. Miller, ed. Inositol Phosphates: Methods and Protocols. Springer US, New York, NY.

**Response 15**
Our study was carried out before the publication of Turner (2020), but is based on the method described in Turner et al. (2012). We have revised the text as follows:

Lines 126-127: The hypobromite oxidation procedure was similar to that reported in Turner (2020).

**Comment 16 (Line 190 and 221)**
Please provide more information on the deconvolution procedure.
Some recent studies appear to have deconvoluted from the baseline to the top of the peaks in the monoester region, which is certain to overestimate the proportion of each signal. This might in turn lead to differences between signals in brominated unbrominated extracts.

**Response 16**
Please see Response 6.

**Comment 17 (Line 262)**
What could the broad signal possibly be, in brominated extracts?

**Response 17**
Please also see Response 8. Furthermore, we speculate that it is a mixture of organic P compounds of complex structure, what could cause steric hindrance, and compounds that contain metal bridges and/or high charge densities, which hinder hypobromite oxidation.

**Comment 18 (Line 225)**
comma instead of period. The persistence of some phosphodiesters suggests

incomplete oxidation.

**Response 18**

We could not find the relevant text that the reviewer is referring to at Line 225 (or elsewhere in the manuscript). We are happy to review this upon advice on the location of the text.

**Comment 19 (Line 276)**
this depends on how spectra were deconvoluted – see point above.

**Response 19**
Please see Response 6.

**Comment 20 (Line 278)**
It's interesting to see evidence for the two conformers of *neo*-IP$_6$. The proportion of the two conformers is definitely related to pH – is it possible that pH was <12 in the extracts, promoting the presence of the two forms?

**Response 20**
Yes, indeed. However, we dissolved the freeze-dried material in 600 μL of 0.25 M NaOH solution, which was spiked with 25 μL of NaOD. We did not measure the pH of the final extract for NMR analysis but the minimal change in the chemical shift of the orthophosphate peak and its location compared to the four *myo*-IP$_6$ peaks suggest that the pH was above 12 (Crouse et al., 2000).

**Comment 21 (Line 283)**
Aren't lower-order esters destroyed by bromination?

**Response 21**
Please see Response 2.

**Comment 22 (Line 292)**
Turner and Richardson 2004 reported signals for two different scyllo-IP4 compounds. Signals from these were not identified in brominated soil extracts, but resolution was not as high as in this study. It looks like only a single scyllo-IP4 isomer was assessed here, so perhaps scyllo-IP4 is underestimated (assuming that the other scyllo-IP4 isomer occurs in soils, and that the tetrakisphosphates resist bromination).

**Response 22**
The reviewer is correct. Obtaining additional standards may increase the detection and amount of lower-order IP in soil extracts. Unfortunately, we were only able to test one *scyllo*-IP$_4$ isomer. This is partly due to limited time and resources, and the rarity of lower-order IP standards. We have revised the manuscript:

Insert (Lines 409-411): Turner and Richardson (2004) reported NMR-signals for two other *scyllo*-IP$_4$ isomers, which could not be tested for in this study due to the lack of available standards.

**Comment 23 (Line 311)**
6 in subscript.

**Response 23**
Corrected.

**Comment 24 (Line 327)**
orthophosphate should increase following bromination, as organic phosphates

are converted to inorganic orthophosphate. This indicates precipitation or loss of phosphates in some other way during the bromination procedure.

**Response 24**
During the hypobromite oxidation, phosphates are precipitated with barium acetate, washed with ethanol and then re-dissolved with ion exchange resins. During these processes, a loss of both, IP and orthophosphate presumably occurs, which we highlight in the manuscript Lines 436-441:

Since the main cause of resistance of IP to hypobromite oxidation is that of steric hindrance, which generally decreases with decreasing phosphorylation state and conformation of the phosphate groups (axial vs. equatorial), we assume that low recoveries of added *myo*-$IP_6$ is due to losses of precipitated $P_{org}$ compounds during the precipitation and dissolution steps. This is supported by the decrease in the concentration of orthophosphate following hypobromite oxidation compared to untreated extracts. Therefore, quantities of IP as reported in the current study should be considered as conservative.

**Comment 25 (Line 404)**
also along the Haast chronosequence: Turner, B. L., A.Wells, and L. M. Condron. 2014. Soil organic phosphorus transformations along a coastal dune chronosequence under New Zealand temperate rain forest. Biogeochemistry 121:595-611. The Baker study on the Franz Josef involved the same sites as Turner et al. 2007, so the separate statement on the Baker study could probably be deleted and the citation rolled into with the others.

**Response 25**
Agreed, we have inserted this citation.

**Comment 26 (Line 418)**
see above. I think the concentrations on the brominated extracts should be considered unreliable, given the apparent loss of phosphorus during the procedure.
It'd probably be better to focus on quantitative values from comparable signals in the unbrominated extracts, and give information from the brominated extracts as qualitative identifications.

**Response 26**
For this reason, we showed both, the concentrations of organic P compounds before and after hypobromite oxidation (Table 4, Table SI1). However, peaks in the phosphomonoester region of untreated extracts have greater overlap, which can affect the accurate quantification of peaks belonging to lower-order IP. Hence, we used the hypobromite oxidation method, which was designed to isolate the IP fraction of soils (Cosgrove and Irving, 1980). Please also see Response 3.

**Comment 27 (Line 434)**
My impression is that the complexity of the monoester region means that deconvolution of all signals could easily account for the apparent broad signal. How does the possibility of more than one compound affect the accuracy of the deconvolution based on a single broad signal?

**Response 27**
Indeed, the findings of McLaren et al. (2019) and our study suggest that the broad signal itself is comprised of several components. These components are taken into account by including the broad signal into the spectral deconvolution fitting procedure (Lines 455-458 in the manuscript). We carried out the $T_2$ relaxation experiment in order to determine if the broad signal itself was comprised of a series of sharp peaks (i.e. inhomogeneous broadening) derived from small molecules, or perhaps a single (or few) peak (i.e.

homogeneous broadening) derived from large and polymeric molecules (Schmidt-Rohr and Spiess, 1994; McLaren et al., 2019). Furthermore, the transverse relaxation time is inversely related to the molecular size, i.e. larger molecules exhibiting shorter $T_2$ times than smaller molecules (Bloembergen et al., 1948; Claridge, 2016). As our results show, the $T_2$ times of the broad signal is significantly shorter compared to the ones of the IP, showing that it is not comprised of many sharp signals as IP but rather few broader signals generated by larger molecules or associations of molecules.

**Comment 28 (Line 436)**
This paragraph is awkward. First, the broad signal is supposedly made up of high molecular weight organic matter, which should be destroyed by bromination. Second, whether the compound forming the broad signal (or compounds, if they exist) occur in the soil is open to question – most scientists working on soil organic matter now accept that much of the high molecular weight material in alkaline soil extracts is formed as an artifact of the extraction procedure. Finally, the statement that the broad signal didn't change after 62 years of cropping seems to indicate precisely the opposite interpretation to that of the authors – that it demonstrates its importance in the soil P cycle. If it's so stable that it never changes, that suggests to me that it's actually fairly unimportant, at least ecologically or agronomically.

**Response 28**
Our hypobromite oxidised NMR spectra showed both, sharp signals and an underlying broad signal fitted with the spectral deconvolution fitting procedure. Because of that, we wanted to test if the broad signal was comprised of many sharp signals generated by small molecules (e.g. IP) or if other, larger molecules were causing the broad signal as reported in McLaren et al. (2015). To test this, we used a 'spin-echo' experiment to determine the transverse relaxation ($T_2$) times of the phosphomonoesters. Our results show that the $T_2$ times of compounds causing the broad signal were different to those of the IP. Therefore, the former are behaving as molecules of apparent high molecular size. Consequently, this broad signal must be taken into account when carrying spectral deconvolution fitting.

The mechanisms for the formation of this phosphomonoester(s) as part of the broad signal are not known. We are not aware of any evidence that shows the broad signal to be an artefact, or that they are formed during the extraction procedure. Our current model appears to be consistent with the organic matter literature. Nebbioso and Piccolo (2011) reported that high molecular weight material of organic matter in soil is an association of smaller organic molecules. These associations however would still cause a broad signal in the phosphomonoester region of soil extracts and could be a reason that some organic molecules containing P are protected from hypobromite oxidation. We have made this clearer in the body text.

Insert Line 458-461: Nebbioso and Piccolo (2011) reported that high molecular weight material of organic matter in soil results from the association of smaller organic molecules. We suggest that these associations would still cause a broad signal in the phosphomonoester region of soil extracts and could be a reason that some organic molecules containing P are protected from hypobromite oxidation.

We consider the compounds causing the broad signal to be important because of two reasons: 1) it exhibits a P pool of considerable amount and unknown structure, whose mobility and potential plant availability (e.g. with certain management strategies) are not known and; 2) the concentrations of more readily available organic P compounds may have been overestimated in the past by attributing the peaks of IP and the broad peak to nucleotides and phospholipid hydrolysis products. Please also see Response 8.

**Comment 29 (Table 3)**
you could combine this table with Table 1 to streamline display items.

**Response 29**

We would prefer not to combine these two tables as Table 1 shows general soil properties not measured in this study and Table 3 focuses on P concentrations based on methods presented in the M&M section. Therefore, we consider Table 3 to be better suited in the Results section.

**Comment 30 (Table 4)**

indicate that the broad peak also represents phosphomonoesters.

**Response 30**

Agreed, we added 'in phosphomonoester region'.

**Comment 31 (Table 5)**

I think it's fairly safe to assume that the *chiro*-IP$_6$ is the D form, given that L-*chiro*-inositol has never been detected in phosphorylated form in nature. Also it's interesting to see from this table that the *neo*+D-*chiro*-IP$_6$ and the majority of the lower order esters were detected only in two of the four soils. I didn't get this impression from reading the text.

**Response 31**

Agreed, we have changed *chiro*-IP$_6$ 2-eq/4-ax to D-*chiro*-IP$_6$ 2-eq/4-ax.

We reported in the Result section 3.3, lines 290-293: *neo*-IP$_6$ was identified in the the 2-equatorial/4-axial and 4-equatorial/2-axial conformations, and c*hiro*-IP$_6$ in the 2-equatorial/4-axial confirmation, of the oxidised extracts in the Cambisol and Gleysol, but were absent in the Ferralsol and the Vertisol (Fig. SI4 and SI5 in the Supporting Information).

To make this clearer in the Discussion section, we inserted (Lines 369-371): In the current study, both conformations could be identified in two of the four soil extracts, which is likely due to improved spectral resolution and sensitivity.

**Comment 32 (Table S1)**

this indicates a considerable proportion of the phosphorus has been lost during the bromination procedure.

**Response 32**

Please see Responses 3, 24 and 26.

**REFERENCES**

Almeida, D. S., Menezes-Blackburn, D., Turner, B. L., Wearing, C., Haygarth, P. M., and Rosolem, C. A.: Urochloa ruziziensis cover crop increases the cycling of soil inositol phosphates, Biology and Fertility of Soils, 54, 935-947, 10.1007/s00374-018-1316-3, 2018.

Bloembergen, N., Purcell, E. M., and Pound, R. V.: Relaxation Effects in Nuclear Magnetic Resonance Absorption, Physical Review, 73, 679-712, 10.1103/PhysRev.73.679, 1948.

Bünemann, E. K., Smernik, R. J., Marschner, P., and McNeill, A. M.: Microbial synthesis of organic and condensed forms of phosphorus in acid and calcareous soils, Soil Biology and Biochemistry, 40, 932-946, https://doi.org/10.1016/j.soilbio.2007.11.012, 2008.

Claridge, T. D. W.: Chapter 2 - Introducing High-Resolution NMR, in: High-Resolution NMR techniques in organic chemistry, 3 ed., edited by: Claridge, T. D. W., Elsevier, Boston, 11-59, 2016.

Cosgrove, D. J., and Irving, G. C. J.: Inositol phosphates : their chemistry, biochemistry and physiology, Studies in organic chemistr, Amsterdam : Elsevier, 1980.

Crouse, D. A., Sierzputowska-Gracz, H., and Mikkelsen, R. L.: Optimization of sample ph and temperature for phosphorus-31 nuclear magnetic resonance spectroscopy of poultry manure extracts, Communications in Soil Science and Plant Analysis, 31, 229-240, 10.1080/00103620009370432, 2000.

Doolette, A. L., and Smernik, R. J.: Facile decomposition of phytate in the solid-state: kinetics and decomposition pathways, Phosphorus, Sulfur, and Silicon and the Related Elements, 193, 192-199, 10.1080/10426507.2017.1416614, 2018.

Golovan, S. P., Meidinger, R. G., Ajakaiye, A., Cottrill, M., Wiederkehr, M. Z., Barney, D. J., Plante, C., Pollard, J. W., Fan, M. Z., Hayes, M. A., Laursen, J., Hjorth, J. P., Hacker, R. R., Phillips, J. P., and Forsberg, C. W.: Pigs expressing salivary phytase produce low-phosphorus manure, Nature Biotechnology, 19, 741-745, 10.1038/90788, 2001.

Irving, G. C. J., and Cosgrove, D. J.: The use of hypobromite oxidation to evaluate two current methods for the estimation of inositol polyphosphates in alkaline extracts of soils, Communications in Soil Science and Plant Analysis, 12, 495-509, 10.1080/00103628109367169, 1981.

Jarosch, K. A., Doolette, A. L., Smernik, R. J., Tamburini, F., Frossard, E., and Bünemann, E. K.: Characterisation of soil organic phosphorus in NaOH-EDTA extracts: a comparison of $^{31}$P NMR spectroscopy and enzyme addition assays, Soil Biology and Biochemistry, 91, 298-309, https://doi.org/10.1016/j.soilbio.2015.09.010, 2015.

Leytem, A. B., Turner, B. L., and Thacker, P. A.: Phosphorus composition of manure from swine fed low-phytate grains, Journal of Environmental Quality, 33, 2380-2383, 10.2134/jeq2004.2380, 2004.

Leytem, A. B., and Maguire, R. O.: Environmental implications of inositol phosphates in animal manures, in: Inositol phosphates: linking agriculture and the environment, edited by: Turner, B. L., Richardson, A. E., and Mullaney, E. J., CABI, Wallingford, 150-168, 2007.

McLaren, T. I., Smernik, R. J., McLaughlin, M. J., McBeath, T. M., Kirby, J. K., Simpson, R. J., Guppy, C. N., Doolette, A. L., and Richardson, A. E.: Complex forms of soil organic phosphorus–A major component of soil phosphorus, Environmental Science & Technology, 49, 13238-13245, 10.1021/acs.est.5b02948, 2015.

McLaren, T. I., Verel, R., and Frossard, E.: The structural composition of soil phosphomonoesters as determined by solution $^{31}$P NMR spectroscopy and transverse relaxation ($T_2$) experiments, Geoderma, 345, 31-37, https://doi.org/10.1016/j.geoderma.2019.03.015, 2019.

Nebbioso, A., and Piccolo, A.: Basis of a Humeomics Science: Chemical Fractionation and Molecular Characterization of Humic Biosuprastructures, Biomacromolecules, 12, 1187-1199, 10.1021/bm101488e, 2011.

Reusser, J. E., Verel, R., Frossard, E., and McLaren, T. I.: Quantitative measures of *myo*-IP$_6$ in soil using solution $^{31}$P NMR spectroscopy and spectral deconvolution fitting including a broad signal, Environmental Science: Processes & Impacts, 22, 1084-1094, 10.1039/C9EM00485H, 2020a.

Reusser, J. E., Verel, R., Frossard, E., and McLaren, T. I.: Quantitative measures of *myo*-IP$_6$ in soil using solution $^{31}$P NMR spectroscopy and spectral deconvolution fitting including a broad signal, Environmental Science: Processes & Impacts, 10.1039/C9EM00485H, 2020b.

Schmidt-Rohr, K., and Spiess, H. W.: Chapter three - High-Resolution NMR techniques for solids, in: Multidimensional Solid-State NMR and Polymers, edited by: Schmidt-Rohr, K., and Spiess, H. W., Academic Press, San Diego, 69-134, 1994.

Smernik, R. J., and Dougherty, W. J.: Identification of phytate in phosphorus-31 nuclear magnetic resonance spectra: the need for spiking, Soil Science Society of America Journal, 71, 1045-1050, 10.2136/sssaj2006.0295, 2007.

Smith, S. E., Read, D. J., and Read, D. J.: Mycorrhizal Symbiosis, Elsevier Science & Technology, San Diego, UNITED KINGDOM, 2008.

Stephens, L. R., and Irvine, R. F.: Stepwise phosphorylation of *myo*-inositol leading to *myo*-inositol hexakisphosphate in Dictyostelium, Nature, 346, 580-583, 10.1038/346580a0, 1990.

Sun, M., Alikhani, J., Massoudieh, A., Greiner, R., and Jaisi, D. P.: Phytate degradation by different phosphohydrolase enzymes: contrasting kinetics, decay rates, pathways, and isotope effects, Soil Science Society of America Journal, 81, 61-75, 10.2136/sssaj2016.07.0219, 2017.

Sun, M., and Jaisi, D. P.: Distribution of inositol phosphates in animal feed grains and excreta: distinctions among isomers and phosphate oxygen isotope compositions, Plant and Soil, 430, 291-305, 10.1007/s11104-018-3723-5, 2018.

Turner, B. L., and Richardson, A. E.: Identification of *scyllo*-inositol phosphates in soil by solution phosphorus-31 nuclear magnetic resonance spectroscopy, Soil Science Society of America Journal, 68, 802-808, 10.2136/sssaj2004.8020, 2004.

Turner, B. L., Richardson, A. E., and Mullaney, E. J.: Inositol phosphates: linking agriculture and the environment, CABI, Wallingford, xi + 288 pp. pp., 2007.

Turner, B. L., Cheesman, A. W., Godage, H. Y., Riley, A. M., and Potter, B. V.: Determination of *neo*- and D-*chiro*-inositol hexakisphosphate in soils by solution $^{31}$P NMR spectroscopy, Environ Sci Technol, 46, 4994-5002, 10.1021/es204446z, 2012.

Turner, B. L.: Isolation of inositol hexakisphosphate from soils by alkaline extraction and hypobromite oxidation, in: Inositol Phosphates: Methods and Protocols, edited by: Miller, G. J., Springer US, New York, NY, 39-46, 2020.

Zhang, X., Li, Z., Yang, H., Liu, D., Cai, G., Li, G., Mo, J., Wang, D., Zhong, C., Wang, H., Sun, Y., Shi, J., Zheng, E., Meng, F., Zhang, M., He, X., Zhou, R., Zhang, J., Huang, M., Zhang, R., Li, N., Fan, M., Yang, J., and Wu, Z.: Novel transgenic pigs with enhanced growth and reduced environmental impact, eLife, 7, e34286, 10.7554/eLife.34286, 2018.

---

## Author Comment (AC2) · 4 May 2020

**REVIEWER REPORT 2**

**Comment 1**
The objective of this manuscript was to characterize and quantify inositol phosphates(IP) in soil extracts following hypobromite oxidation using $^{31}$P nuclear magnetic resonance (P NMR) spectroscopy. This is a very technical paper with respect to the chemical methods utilized. Given that the mandate of this journal is: "interactions between the biological, chemical, and physical processes in terrestrial or extra-terrestrial life with the geosphere, hydrosphere, and atmosphere. The objective of the journal is to cut across the boundaries of established sciences and achieve an interdisciplinary view of these interactions" (from the journal website)", this paper does not seem like a good fit for the journal. While the authors identified a wide range of different P compounds in their four soil samples, no attempt was made to relate these compounds back to broader biological, chemical or physical processes within these soils. As such, it will not be of interest to the majority of Biogeosciences readers, as currently written, and will likely be overlooked by the scientists who would be interested in such a technical paper. In my opinion, this would be a better fit in either an environmental chemistry journal or in the chemistry section of a soil science journal. Thus, in my opinion the authors should withdraw this paper from this journal and submit it to another journal that better fits the paper's focus. If the authors choose not to do this, then they must significantly revise the manuscript to keep it within the journal's scope, to clearly demonstrate the significance of these identified P compounds to P cycling in these soils, and to P cycling more broadly.

**Response 1**
Inositol phosphates are a very important component of the P cycle in both agricultural and environmental contexts. Indeed, several recent reviews have highlighted a stagnation of advancing our knowledge of the P cycle to address global challenges due to a lack of knowledge on organic P (George et al., 2018; Haygarth et al., 2018; McLaren et al., 2020). Our study provides new information on the chemical nature of a multitude of organic P species, which is essential to understand processes relating to their flux in nature and their function in the soil system. Furthermore, the production/accumulation as well as the hydrolysis of IP to lower order IP, involves the cycling of P and C in soil.

We used a novel approach of combining chemical extraction, hypobromite oxidation, and multiple NMR techniques to better understand the chemical composition of soil organic P. Furthermore, we strongly believe that our publication will be of great interest to a broad audience, including scientists working in agriculture, environment, sediments and waters. Lastly, we highlight that our study is the first to report the existence of 11 inositol phosphate species using direct spectroscopic evidence, and also provide new insight on the chemical and structural composition of 'complex' phosphomonoesters. We also thank the reviewer for their positive comment of the paper later in their review (see Comment 28).

**Comment 2**
Abstract: As written, the abstract make it clear that this is chemistry methods paper, not a biogeochemical study, because the results and conclusions highlighted in the abstract indication only that the authors were able to identify these peaks, but make no reference to their relative importance in the studied soils and to P cycling in these and other soils. This supports my point above that this is not an appropriate journal for this paper as currently written.

**Response 2**
In addition to Response 1 of Reviewer 2, we highlight the discussion in the body text on the importance and implications of our results (lines 24-26 in the Abstract, lines 355-359, 379-380, 388-402, 429-434, 462-468 in the Discussion section and lines 492-497 in the Conclusion section).

**Comment 3**

In addition the abstract needs to be more carefully edited, as it is awkwardly written in places. For example, lines 14-15: "include the A horizon of a Ferrasol from Columbia, of a Cambisol from Switzerland, of a Gleysol from Switzerland and of a Cambisol from Germany" should be "include A horizons from a Ferrosol(Columbia), a Cambisol and a Gleysol froom Switzerland, and a Cambisol from Germany".

**Response 3**

Agreed, we have reworded the sentence:

Changed from (lines 14-15): Soil samples analysed include the A horizon of a Ferralsol from Colombia, of a Cambisol from Switzerland, of a Gleysol from Switzerland and of a Cambisol from Germany.

Changed to (lines 14-15): Soil samples analysed include A horizons from a Ferralsol (Colombia), a Cambisol and a Gleysol from Switzerland, and a Cambisol from Germany.

**Comment 4**

And why is the phrase "(using solution 31P NMR spectroscopy)" included inline 19, given that the method was given in line 13?

**Response 4**

We have deleted "(using solution 31P NMR spectroscopy)".

**Comment 5**

Introduction: The introduction provides a good overview of the chemical methodology for extracting and characterizing IP in soil, as would be expected for a chemical methods paper. It gives a very brief overview of the factors generally controlling IP in soils, but doesn't give much information about why there is a need to specifically characterize all of these different IP forms. What insights into soil P cycling would we gain from identifying these compounds that we don't already have by from the IP compounds we can already identify?

**Response 5**

Please see Response 1 of Reviewer 2. In addition, the majority of NMR studies have identified a small selection of compounds in the phosphomonoester region of NMR spectra on soil extracts (McLaren et al 2020). These are typically four $IP_6$ compounds, α- and β-glycerophosphate, and some RNA mononucleotides. Consequently, most studies have focused on the cycling of $IP_6$, which is considered relatively stable in soil. In the current study, we report up to 70 sharp signals in the phosphomonoester region of NMR spectra on soil extracts following hypobromite oxidation, which is considerably more than that typically reported in the literature. We could identify on average 48% of peaks in this region as arising from inositol phosphates, however, it is likely a much greater proportion of these sharp peaks will be due to inositol phosphates due to their resistance to hypobromite oxidation.

The majority of organic P studies have focused on the cycling of $IP_6$, particularly of *myo*-$IP_6$ (McLaren et al. 2020). We show that there is a much greater diversity of organic P compounds than previously thought, and second that they appear to be predominately lower-order inositol phosphates. This has major consequences to our understanding of P cycling, given the different mechanisms and compounds involved than previously thought and their unknown function in the soil system.

Insert Lines 95-98: We hypothesise that a large portion of sharp peaks in the phosphomonoester region of untreated soil extracts would be resistant to hypobromite oxidation, which would indicate the presence of a wide variety of IP. This would have major

consequences to our understanding of P cycling in terrestrial (and aquatic) ecosystems, as much more organic P compounds and mechanisms would be involved than previously thought.

**Comment 6**
And what information would be expected from analyzing them in different soils?

**Response 6**
A diverse set of soils provides the opportunity to identify a greater array of organic P species than what might be present in only one soil. The diversity of soil properties may also reveal different relative contributions of organic P species than that present in a particular soil type.

**Comment 7**
And the hypothesis seems to be something that was tacked on at the end, and doesn't make a lot of sense: "We hypothesize that a large portion of sharp peaks in the phosphomonoester region of untreated soil extracts would be resistant to hypobromite oxidation, which would indicate the presence of IP". This again emphasizes that this is a chemical methods paper only.

**Response 7**
Please see Response 5 of Reviewer 2. In addition, in a recent paper we obtained high-resolution NMR spectra that exhibited a plethora of sharp peaks and an underlying broad peak in the phosphomonoester region on soil extracts (Reusser et al 2020). This suggested a much greater diversity of organic P species than previously thought. The identity of these sharp peaks was largely unknown and could not be attributed to the limited number of RNA mononucleotides and two glycerophosphates often reported in the literature. Furthermore, a review of the literature from the 1950s to 1970s indicated some studies report the presence of lower-order inositol phosphates in soil extracts using chromatographic approaches. Consequently, we hypothesised that a large portion of sharp peaks in the phosphomonoester region of untreated soil extracts would be resistant to hypobromite oxidation, which would indicate the presence of inositol phosphates. If the majority of sharp peaks disappeared following hypobromite oxidation, then this would indicate that the sharp signals were due to non-inositol phosphate compounds. We combined previously published methods to test this hypothesis, but did not seek to advance or test the efficacy of these methods as is typically done in a 'methods' paper.

**Comment 8**
Other points in the Introduction: l. 35: "Riley Andrew et al., 2006)" why is the authors first name included (Andrew M. Riley is the first author of the paper)? This should be "Riley et al., 2006". And the listing in the References (l. 641-644) contains the first names of other authors of this paper. "Shears Stephen, B" should be "Shears, SB", and "Potter Barry VL" should be "Potter BVL". The correct names are very obvious when reading the manuscript, so I'm not sure why they are incorrect here.

**Response 8**
Agreed, we changed the reference accordingly. This error occurred because of a formatting issue in the EndNote library.

Reference entry changed to (lines 682-685): Riley, A. M., Trusselle, M., Kuad, P., Borkovec, M., Cho, J., Choi, J. H., Qian, X., Shears, S. B., Spiess, B., and Potter, B. V. L.: *scyllo*-Inositol pentakisphosphate as an analogue of *myo*-inositol 1,3,4,5,6-pentakisphosphate: Chemical synthesis, physicochemistry and biological applications, ChemBioChem, 7, 1114-1122, 10.1002/cbic.200600037, 2006.

**Comment 9**
l. 39 and elsewhere in the text: when citing a list of references, it is conventional to list them in order from oldest to most recent.

**Response 9**
We have updated the reference list.

**Comment 10**
l. 87: "was resistant" should be "were resistant", because it modified "signals", which is plural.

**Response 10**
Corrected.

**Comment 11**
Methods: As written, there is far too much technical information (e.g. about the transverse relaxation experiments), which will not be of any interest to the majority of readers of this journal.

**Response 11**
We are happy to reduce this if requested by the Editor. However, the approach is not well known outside of the NMR and organic P communities, and the additional information may be useful for understanding and for reproducibility in future experiments.

**Comment 12**
And other important information seems to be missing. See specific points listed below. Also, I believe that Turner has published a new paper of the hypobromite oxidation method. How does the method used compare to that method.

**Response 12**
We carried out the hypobromite oxidation procedure based on the method of Turner et al. (2012), and prior to the publication of Turner (2020). Briefly, Turner et al (2020) suggest taking a 10 mL aliquot of soil extract, adding 2 g of NaOH, and then adding 0.5 mL of bromine. This is slightly different to that reported in Turner et al (2012). In the current study, we similarly take a 10 mL aliquot of soil extract, but add 1 mL of 10 M NaOH, and add 0.6 mL of bromine. Please see Response 3 of Reviewer 1.

**Comment 13**
l. 117: Please provide information on the total volume of extractant used and the total volume of filtrate produced, to help the reader put the hypobromite oxidation experiments into context. In line 121, it indicates that "10 mL of the filtrate was used". What proportion of the total filtrate is this – 10% or 100%?

**Response 13**
We used 25% of the total filtrate for the hypobromite oxidation. We have made this clearer in the manuscript:

Inserted (lines 121-123): Concentrations of organic P for NMR analysis were carried out using the NaOH-EDTA extraction technique of Cade-Menun et al. (2002) at a soil to solution ratio of 1:10, i.e. extracting 4 g of soil with 40 mL of extractant.

Changed from (lines 121-123): Briefly, 10 mL of the filtrate was placed in a three necked round bottom flask equipped with a septum, a condenser, a magnetic stir bar and thermometer (through a claisen adapter with $N_2$ adapter).

Changed to (lines 127-129): Briefly, 10 mL of the NaOH-EDTA filtrate (section 2.2) was placed in a three necked round bottom flask equipped with a septum, a condenser, a magnetic stir bar and thermometer (through a claisen adapter with $N_2$ adapter).

**Comment 14**

l. 144-145: This sentence is awkwardly written. Change "...in solution is that of molybdate unreactive P (MUP), which is considered to be largely that of organic P" to "in solution is molybdate unreactive P (MUP), which is predominantly organic P for these samples"

**Response 14**

Agreed.

Changed from (lines 144-145): The difference in concentrations of total P and MRP in solution is that of molybdate unreactive P (MUP), which is considered to be largely that of organic P.

Changed to (lines 150-151): The difference in concentrations of total P and MRP in solution is molybdate unreactive P (MUP), which is predominantly organic P for these samples.

**Comment 15**

l. 146-147: "a duplicate sample of the Cambisol and the Gleysol was spiked" should be "duplicate samples of the Cambisol and Gleysol were spiked"

**Response 15**

Corrected.

**Comment 16**

l. 161-162: The inclusion of the Vestergren et al. 2012 paper here confused me. This group left their samples to sit overnight because they used a sulfide treatment to remove paramagnetic ions. Was this also done for the current study? If so, then please describe the sulfide treatment more clearly. If not, then it would be better to replace this reference with one that is more appropriate.

**Response 16**

Vestergren et al. (2012) report in their body text: "Extraction of soils with NaOH/EDTA is known to hydrolyze several forms of phosphodiesters. This is considered an unavoidable drawback of the method, but it has been pointed out that it does not exclude deriving the original P composition when hydrolysis products can be traced back.[21] Therefore, when a hydrolysis product is observed, it must be determined what fraction of the compound was originally present in the soil, versus formed during extraction.[19] Whereas the longer sample preparation time for sulfide treatment increases hydrolysis (Figure S3 of the Supporting Information), the 2D methodology is very well suited to trace observed compounds back to their precursors". The citation of Vestergren et al. (2012) in our manuscript refers to their findings in the Supporting Information (Figure 3). The authors present NMR spectra and report that more hydrolysis of phosphodiesters are due to the "longer exposure to high pH", and that the 'resting' time of the extracts in the study was 18-20 hours at room temperature. The authors note in their study the mechanism of alkaline hydrolysis of organic P compounds to their hydrolysis products and the necessity of a reaction period lasting several hours for sufficient hydrolysis.

**Comment 17**

l. 193-195: Something seems to be missing here for the measurement of N observability. Using Ptot ICP-OES only makes sense if the entire sample after freeze-drying was used for the NMR analysis. However, that does not seem to be the case for this study. While it appears that the total mass of lyophilized material was used for the brominated samples (l. 167-168), a set mass (120 g) of the non-brominated lyophilized material was used, with no indication of how much of the total lyophilized material this represents. The proportion of total mass used must be factored into the equation to correctly determine NMR observability. This

would also explain the differences in observability reported in the supplementary information (SI) for the brominated and unbrominated samples.

**Response 17**

$P_{tot}$ NMR and $P_{tot}$ ICP-OES refer to the P concentrations in mg P per kg soil measured in the extracts. Hence, the analysed P contents in the extracts were back-calculated to the original concentrations in the soil, including any partitioning in the extraction, freeze-drying and re-dissolving processes. We made this clearer in the text by inserting the units of the two parameters.

Insert (lines 206-208): ,where $P_{tot}$ NMR refers to the total P content in mg P/kg$_{soil}$ detected in the soil extracts using solution $^{31}P$ NMR spectroscopy and $P_{tot}$ ICP-OES refers to the total P concentration in mg P/kg$_{soil}$ measured in the soil extracts prior to freeze-drying using ICP-OES.

**Comment 18**

l. 206-225: There is no need to include this much detail about the transverse relaxation papers. As noted above, the majority of readers of this paper in this journal will not be interested in these details. In addition, this appears to be a repeat of what was done for the McLaren et al. 2019 study. As such, all that is needed is to cite the previous publication. If the authors really thing this much detail is needed, it could be included in the SI.

**Response 18**

Please see Response 11.

**Comment 19**

L. 226-233: Why are methods for statistical analyses reported here, when no results of statistical analysis are included in the Results, Discussion or SI?

**Response 19**

We report in our studies average values as well as standard deviations. Furthermore, we carried out the one-way ANOVA with subsequent multi comparison of mean values using the Tukey's significance honestly significant difference procedure to determine whether the $T_2$ of the broad peak was significantly different from the IP peaks. The result of this statistical analysis is reported in the text, lines 330-332: The average (n=4) $T_2$ times of the broad peak was significantly different than that of *scyllo*- and *myo*-IP$_6$ ($p < 0.05$).

**Comment 20**

Results: 1. Please provide spectra showing the entire spectrum for each brominated and unbrominated sample, scaled to allow the reader to see the full height of orthophosphate and the relative heights of other peaks compared to orthophosphate. All of the spectra currently in the manuscript show the monoester region only, with the orthophosphate peak truncated. This is needed to get a full sense of all the peaks for each sample, especially for the brominated samples.

**Response 20**

The main reaction was oxidation, not bromination of the samples. The aim of our study was the identification of IP, whose peaks appear in the phosphomonoester region. Hence, our spectra focus on the phosphomonoester region, which is also where the majority (> 99%) of NMR signals are located. We are unsure why the inclusion of the whole spectrum would add to the information already provided in Table 4. Nevertheless, we are willing to add the spectra of the Gleysol and Cambisol (Figure 2), where considerable amounts of phosphodiesters were measured before hypobromite oxidation, to the supporting information.

[Figure]

**Figure 2.** Solution $^{31}P$ nuclear magnetic resonance (NMR) spectra (500 MHz) of the orthophosphate, phosphomonoester and phosphodiester region on untreated (UT, on top) and hypobromite oxidised (HO, below) 0.25 M NaOH + 0.05 M EDTA soil extracts of the Gleysol (right) and Cambisol (left). Signal intensities were normalised to the MDP peak intensity. The vertical axes were increased for improved visibility of spectral features, as indicated by a factor.

**Comment 21**
The usefulness of the spectra shown in Fig. 3 are not clear. I am used to looking at NMR spectra, and I found these confusing, as with the exception of the Gleysol the red lines show little but noise. Again, this would be appropriate for a chemical methods paper, emphasizing that this is not the best journal for this study.

**Response 21**
The aim of the transverse relaxation ($T_2$) experiments was to determine if the underlying broad signal itself is caused by sharp peaks of IP or if another compound of larger structure than IP resisted hypobromite oxidation (Please also see Response 28, Reviewer 1). The red line of Figure 3 does not show a normal NMR soil spectrum but the result of the $T_2$ experiment with the longest spin-echo delay for each hypobromite oxidised soil sample. The spectra in black with the shortest spin-echo delay can be interpreted as a "normal" NMR soil spectra. We applied increasing spin-echo delays and acquired the resulting spectra for each step. However, due to visibility reasons, we only show the results of the shortest (black) and longest spin-echo delay (red). This presentation is normal for $T_2$ experiments (Claridge, 2016; Li et al., 2018a).

Figure 3 shows that the sharp peaks of IP after a spin-echo delay of 80*τ are still present (red line). In contrast, the broad peak partially disappears along with the orthophosphate peak, showing nothing else than noise. This highlights that the broad peak and orthophosphate peak are not of the same chemical composition as the rest of the sharp peaks, as it would not be visible only in the black spectra. As the $T_2$ are inversely related to a compound's molecular size, our results support the findings of Jarosch et al. (2015) and McLaren et al. (2015b); McLaren et al. (2019) that the compounds causing the broad signal are of larger molecular size than IP.

**Comment 22**
I am concerned that the authors report signals for non-IP compounds in their brominated spectra. In my experience with this technique, if there are any peaks for non-IP compounds, that suggests that the oxidation was incomplete. And that in turn raises questions about the authors' assignment of peaks in the brominated samples. How confident are the authors that all of the peaks were present in their soils prior to extraction and hypobromite oxidation? Isn't it possible that bromination degraded some high IPs (e.g. $IP_6$) to lower IPs ($IP_5$ and $IP_4$)? The recovery of the added *myo*-$IP_6$ was only 20 and 47%, which suggests it may have been degraded.

**Response 22**
According to the method, inositol hexakisphosphates and pentakisphosphates are stable to hypobromite oxidation, please see Response 2 of Reviewer 1. We tested the oxidation efficacy in a pilot study (Response 3, Reviewer 1). Furthermore, bromine was added in excess. If not all organic P species have been oxidised, this suggests that they are stable to hypobromite oxidation, highlighting their chemical stability. The losses occurred most certainly during the precipitation and re-dissolving procedure and not because of degradation. Please also see Response 24 of Reviewer 1. Furthermore, we identified inositol pentakisphosphates in untreated extracts, lines 312-320.

**Comment 23**
l. 255: change "Although," to "However,"

**Response 23**
Corrected.

**Comment 24**
l. 273: "A detailed view of the phosphomonoester region of spiked extracts is shown" should be "Detailed views of the phosphomonoester regions of spiked samples are shown"

**Response 24**
Agreed.

Changed from (lines 273-274): A detailed view of the phosphomonoester region of spiked extracts is shown in Fig. SI1 to SI5 of the Supporting Information.

Changed to (lines 286-287): Detailed views of the phosphomonoester regions of spiked samples are shown in Fig. SI1 to SI5 of the Supporting Information.

**Comment 25**
l. 306-316: I do not see the need to include any of this information about spin-echo analysis of selected P compounds in the current paper, as it will not be of any interest to the majority of readers of this paper in this journal.

**Response 25**
The 'spin-echo' analysis was carried out to provide evidence that there were other compounds different to IP resistant to hypobromite oxidation. Without these results, one

could assume that the broad signal itself could be comprised of sharp peaks caused by IP. Please also see Response 27, Review 1.

**Comment 26**
Discussion: The P-NMR literature cited in this section seems biased to papers by the Smernik group. I have concerns about this because that group prepared their samples for NMR differently from most other groups, and from what was done for the current study. As such, results from that group may not be directly comparable here.

**Response 26**
We are unsure what the reviewer means by their comment regarding citations. Citations are primarily used to support the claims of the authors made in the body text. If the reviewer believes we have incorrectly used a citation when supporting a claim, then we are happy to make corrections. Unfortunately, the reviewer has not provided any evidence to support her or his claim.

We are unsure what the reviewer means by this comment regarding NMR sample preparation. A comparison of methods for preparing NMR samples by Dr Ronald Smernik (e.g. Smernik and Dougherty (2007)) and that reported in the current study, clearly shows a large difference in sample preparation. Both of these methods also slightly differ to other groups using NMR approaches (Cade-Menun and Liu, 2014). Indeed, our approach is based on the studies of Vincent et al. (2013) and Spain et al. (2018), which is optimised to the high-resolution NMR spectrometers we have access to.

Lastly, we note that McLaren et al. (2019) is the only study reporting transverse-relaxation ($T_2$) experiments for organic P compounds in soil mineral samples. In addition, studies by Smernik et al. have also done much work on identifying lower-order IP in plant samples using solution $^{31}P$ NMR spectroscopy.

**Comment 27**
In addition, it shows an unfamiliarity with the broader P-NMR literature, which is of concern.

**Response 27**
Please see Response 26. In addition, we are unsure why the reviewer has made this assertion given the recent review paper on the chemical nature of soil organic P by two of the co-authors (McLaren et al. (2020)). Of course, it is possible that we may have made an error and have missed a relevant study. In this case, we would be happy to make corrections and strengthen the claims already made in the text. Unfortunately, the reviewer has not provided any details where a publication might have been missed or incorrectly cited.

**Comment 28**
In general, however, I think the authors have done a reasonable job of trying to relate these P compounds to the literature and to the soils, which would be suitable to this journal. However, they should note the overall small proportion of total P that some of these compounds comprise. Are compounds in such low concentrations really an integral component of P cycling.

**Response 28**
We thank the reviewer for the positive comment.

For example, water extractable inorganic P can be very small in terms of concentration but rather important in terms of function. In addition, we note that total IP comprised up to 18% of total $P_{org}$ in hypobromite oxidised extracts and compounds causing the broad signal on average 23% of total $P_{org}$ in untreated extracts. In our opinion, these organic P pool should not be neglected. Furthermore, ratios of $IP_6$ to $IP_5$ could provide a tool for assessing stability

of IP in soil systems, please see Response 2. water extractable inorganic P can be very small in terms of concentration but rather important functionally.

**Comment 29**
And in my opinion, section 4.3 is not appropriate for this journal and would not be of interest to the majority of readers, and so should be cut.

**Response 29**
This section refers to the structural composition and possible stability of compounds causing the broad signal in soil, which has implications to our understanding of soil organic matter and 'legacy' P in agroecosystems. Lines 462-468 in the manuscript: Since a portion of the broad signal is resistant to hypobromite oxidation, this suggests the organic P is complex and in the form of polymeric structures. The chemical resistance of the broad signal to hypobromite oxidation may also indicate a high stability in soil (Jarosch et al., 2015). Annaheim et al. (2015) found that concentrations of the broad signal remained unchanged between three different organic fertiliser strategies after 62 years of cropping. In contrast, the organic P compounds annually added with the fertilisers were completely transformed or lost in the slightly acidic topsoil of the field trial. Nebbioso and Piccolo (2011) reported that high molecular weight material of organic matter in soil is an association of smaller organic molecules. These associations however would still cause a broad signal in the phosphomonoester region of soil extracts and could be a reason that some organic molecules containing P are protected from hypobromite oxidation. The large proportion of the broad signal in the total organic P pool demonstrates its importance in the soil P cycle.

**Comment 30**
l. 322-324: Other studies have looked at what was not extracted by NaOH-EDTA, including with acid extraction after NaOH-EDTA or with solid-state P-NMR. See for example studies by He et al. These would be more appropriate to cite here than McLaren et al., 2015a

**Response 30**
It is unclear which particular study by He et al the reviewer is referring to. McLaren et al. (2015a) determined the total concentrations of soil P using X-ray fluorescence spectroscopy, which was similarly the case here. The authors then compared these measures with that of aqua regia digestion, the ignition-$H_2SO_4$ and NaOH-EDTA extraction techniques, and also the summation of P fractions from a sequential chemical fractionation procedure based on Hedley et al. (1982). The authors report that the native soil of their study contained a fraction of strongly-held mineral P that was neither acid nor alkali extractable. They also considered the XRF method to be the most reliable for quantifying concentrations of total P in soil, which was similar to the summation of P fractions by sequential chemical fractionation. Furthermore, the authors provide supporting evidence that a relatively small portion of alkaline soluble organic P was not extracted by NaOH-EDTA.

We report in our study, Lines 337-340: On average, 44 % of total P (as measured with XRF) was extracted by NaOH-EDTA, which is consistent with previous studies (Turner, 2008; Li et al., 2018b; McLaren et al., 2019). The non-extractable pool of P is likely to comprise of inorganic P as part of insoluble mineral phases, but could also contain some organic P (McLaren et al., 2015a). Hence, we refer to the pool of P not extracted by NaOH-EDTA but measured by XRF. Therefore, we consider the publication of McLaren et al. (2015a) as the most suiable in this context.

The reviewer could be referring to He et al. (2007). Here the authors reported that P recoveries in NaOH-EDTA extracts of poultry manure were lower compared to extracts of dairy manure. The authors attributed this lower recovery to the higher Ca content in the poultry manure. Increased Ca in the poultry manure may have resulted in less soluble forms of P that were not extracted with NaOH-EDTA. By using an additional extraction step (1 M

HCl) following the NaOH-EDTA step, the authors were able to recover the remaining P from the poultry manure. Furthermore, solution $^{31}$P NMR spectra of the HCl extract revealed that the majority of P was present as orthophosphate and to a lesser extent phytate. However, the study of He et al. (2007) was carried out on manure samples and are not relevant to soil samples.

**Comment 31**
l. 333-334: "This will result in the production of carbon dioxide and simple organic acids" This sentence does not seem to be relevant here. How is this related to P?

**Response 31**
It relates to what happens to the organic molecules containing phosphate as functional group. It gives more detail on what actually happens to the organic molecules during the hypobromite oxidation procedure. We reworded the sentence to make this clearer.

Changed from (lines 333-334): This will result in the production of carbon dioxide and simple organic acids.

Changed to (lines 349-351): The products of hypobromite oxidation are most probably carbon dioxide, simple organic acids from the oxidative cleavage of the phosphoesters and orthophosphate.

**Comment 32**
l. 340-342: If the authors had not shown peaks other than monoesters and orthophosphate, I might agree with them that the peaks in the monoester region are all IP. However, it is clear from the results they have shown that they did not have complete oxidation of all P compounds. So how can they be confident that they only have IP in the monoester region? This must be addressed.

**Response 32**
Please see Responses 3 and 5 of Reviewer 1.

**Comment 33**
l. 348-350: I'm confused by the some of the papers cited here. Why are studies that did not use chromatography cited here to make a point about chromatography. Please rephrase, or remove the non-chromatography references.

**Response 33**
It appears the reviewer has misread the sentence. We provide two different citation groups for studies involving chromatography and NMR spectroscopy (see below).

Lines 365-367: The detection of *myo*-, *scyllo*-, *chiro*, and *neo*-IP$_6$ in untreated and hypobromite oxidised soil extracts is consistent with previous studies using chromatography (Irving and Cosgrove, 1982; Almeida et al., 2018) and NMR (Turner and Richardson, 2004; McLaren et al., 2015b; Jarosch et al., 2015; Vincent et al., 2013; Doolette et al., 2011a).

**Comment 34**
l. 356-363: As noted above, the authors did not have compete oxidation of all non-IP compounds in their extracts. So how can they be certain that this peak at 4.36 is an IP compound and not α-glycerol. In addition, other groups have reported a peak that sits very close to α-glycerol, and have urged caution about identifying this peak without spiking. This emphasizes a need for a broader review of the literature than just papers from the Smernik group.

**Response 34**

We can confirm that bromine was present in excess and that soil extracts were kept at reflux following bromine addition. Furthermore, the volume of bromine added relative to the aliquot of soil extract was similar or greater in our study compared to that in previous studies (Turner et al 2012; Turner & Richardson 2004). Please see Responses 3 and 5 of Reviewer 1.

Unfortunately, the reviewer has not provided the reference to support his or her claim. We are not aware of any study that has identified another organic P species at the chemical shift at or near that of α-glycerophosphate. Nevertheless, in the current study, the assignment of α-glycerophosphate was based on spiking experiments in untreated soil extracts. Following hypobromite oxidation, this peak disappeared, revealing two peaks belonging to IP. This then provided strong evidence that the peak originally assigned to α-glycerophosphate was in fact due to an IP.

The assignment of one of the aforementioned peaks in hypobromite extracts was confirmed by spiking experiments with *neo*-$IP_6$ in the 2-equatorial/4-axial conformation. This resulted in the increased peak intensity at 4.37 ppm (C2,5) and its corresponding peak at 4.11 ppm (C1,3,4,6), which occurred at the known peak ratio of 4:2 for *neo*-$IP_6$ in the 2-equatorial/4-axial conformation, see Figure SI4 with the spiking results. Consequently, our results highlight the need for caution when assigning the α-glycerophosphate peak based on spiking experiments alone with α-glycerophosphate in untreated soil extracts. We would recommend that spiking with *neo*-$IP_6$ would also occur. We have revised the text, lines 376-380: Whilst a peak at δ 4.36 ppm would be assigned to α-glycerophosphate based on spiking experiments in the untreated extracts of the Cambisol and the Gleysol, hypobromite oxidation revealed the presence of the 2-equatorial/4-axial C2,5 peak of *neo*-$IP_6$ at δ 4.37 ppm, and also an unidentified peak at δ 4.36 ppm in the Cambisol. Therefore, the assignment and concentration of α-glycerophosphate may be unreliable in some soils of previous studies.

**Comment 35**

l. 370: change "extracts, which the" to "extracts, of which the"

**Response 35**

Corrected.

**Comment 36**

l. 383: add spaces between the numbers and words here: "1axial" should be "1 axial" or "1-axial", etc.

**Response 36**

Agreed.

Changed from (line 383): the 1axial/5equatorial and 5axial/1 equatorial forms of *myo*-(1,2,3,4,6)-$IP_5$ are in a dynamic equilibrium,

Changed to (lines 403-405): the 1-axial/5-equatorial and 5-axial/1-equatorial forms of *myo*-(1,2,3,4,6)-$IP_5$ are in a dynamic equilibrium,

**REFERENCES**

Annaheim, K. E., Doolette, A. L., Smernik, R. J., Mayer, J., Oberson, A., Frossard, E., and Bünemann, E. K.: Long-term addition of organic fertilizers has little effect on soil organic phosphorus as characterized by $^{31}$P NMR spectroscopy and enzyme additions, Geoderma, 257-258, 67-77, https://doi.org/10.1016/j.geoderma.2015.01.014, 2015.

Cade-Menun, B., and Liu, C. W.: Solution phosphorus-31 nuclear magnetic resonance spectroscopy of soils from 2005 to 2013: a review of sample preparation and experimental parameters, Soil Science Society of America Journal, 78, 19-37, 10.2136/sssaj2013.05.0187dgs, 2014.

Cade-Menun, B. J., Liu, C. W., Nunlist, R., and McColl, J. G.: Soil and litter phosphorus-31 nuclear magnetic resonance spectroscopy, Journal of Environmental Quality, 31, 457-465, 10.2134/jeq2002.4570, 2002.

Claridge, T. D. W.: Chapter 2 - Introducing High-Resolution NMR, in: High-Resolution NMR techniques in organic chemistry, 3 ed., edited by: Claridge, T. D. W., Elsevier, Boston, 11-59, 2016.

George, T. S., Giles, C. D., Menezes-Blackburn, D., Condron, L. M., Gama-Rodrigues, A. C., Jaisi, D., Lang, F., Neal, A. L., Stutter, M. I., Almeida, D. S., Bol, R., Cabugao, K. G., Celi, L., Cotner, J. B., Feng, G., Goll, D. S., Hallama, M., Krueger, J., Plassard, C., Rosling, A., Darch, T., Fraser, T., Giesler, R., Richardson, A. E., Tamburini, F., Shand, C. A., Lumsdon, D. G., Zhang, H., Blackwell, M. S. A., Wearing, C., Mezeli, M. M., Almås, Å. R., Audette, Y., Bertrand, I., Beyhaut, E., Boitt, G., Bradshaw, N., Brearley, C. A., Bruulsema, T. W., Ciais, P., Cozzolino, V., Duran, P. C., Mora, M. L., de Menezes, A. B., Dodd, R. J., Dunfield, K., Engl, C., Frazão, J. J., Garland, G., González Jiménez, J. L., Graca, J., Granger, S. J., Harrison, A. F., Heuck, C., Hou, E. Q., Johnes, P. J., Kaiser, K., Kjær, H. A., Klumpp, E., Lamb, A. L., Macintosh, K. A., Mackay, E. B., McGrath, J., McIntyre, C., McLaren, T., Mészáros, E., Missong, A., Mooshammer, M., Negrón, C. P., Nelson, L. A., Pfahler, V., Poblete-Grant, P., Randall, M., Seguel, A., Seth, K., Smith, A. C., Smits, M. M., Sobarzo, J. A., Spohn, M., Tawaraya, K., Tibbett, M., Voroney, P., Wallander, H., Wang, L., Wasaki, J., and Haygarth, P. M.: Organic phosphorus in the terrestrial environment: a perspective on the state of the art and future priorities, Plant and Soil, 427, 191-208, 10.1007/s11104-017-3391-x, 2018.

Haygarth, P. M., Harrison, A. F., and Turner, B. L.: On the history and future of soil organic phosphorus research: a critique across three generations, European Journal of Soil Science, 69, 86-94, 10.1111/ejss.12517, 2018.

He, Z., Cade-Menun, B. J., Toor, G. S., Fortuna, A.-M., Honeycutt, C. W., and Sims, J. T.: Comparison of phosphorus forms in wet and dried animal manures by solution phosphorus-31 nuclear magnetic resonance spectroscopy and enzymatic hydrolysis, Journal of Environmental Quality, 36, 1086-1095, 10.2134/jeq2006.0549, 2007.

Hedley, M. J., Stewart, J. W. B., and Chauhan, B. S.: Changes in Inorganic and Organic Soil Phosphorus Fractions Induced by Cultivation Practices and by Laboratory Incubations1, Soil Science Society of America Journal, 46, 970-976, 10.2136/sssaj1982.03615995004600050017x, 1982.

Jarosch, K. A., Doolette, A. L., Smernik, R. J., Tamburini, F., Frossard, E., and Bünemann, E. K.: Characterisation of soil organic phosphorus in NaOH-EDTA extracts: a comparison of $^{31}$P NMR spectroscopy and enzyme addition assays, Soil Biology and Biochemistry, 91, 298-309, https://doi.org/10.1016/j.soilbio.2015.09.010, 2015.

Li, L., Zhang, M., Bhandari, B., and Zhou, L.: LF-NMR online detection of water dynamics in apple cubes during microwave vacuum drying, Drying Technology, 36, 2006-2015, 10.1080/07373937.2018.1432643, 2018a.

Li, M., Cozzolino, V., Mazzei, P., Drosos, M., Monda, H., Hu, Z., and Piccolo, A.: Effects of microbial bioeffectors and P amendments on P forms in a maize cropped soil as evaluated by $^{31}$P–NMR spectroscopy, Plant and Soil, 427, 87-104, 10.1007/s11104-017-3405-8, 2018b.

McLaren, T. I., Simpson, R. J., McLaughlin, M. J., Smernik, R. J., McBeath, T. M., Guppy, C. N., and Richardson, A. E.: An assessment of various measures of soil phosphorus and the

net accumulation of phosphorus in fertilized soils under pasture, Journal of Plant Nutrition and Soil Science, 178, 543-554, 10.1002/jpln.201400657, 2015a.

McLaren, T. I., Smernik, R. J., McLaughlin, M. J., McBeath, T. M., Kirby, J. K., Simpson, R. J., Guppy, C. N., Doolette, A. L., and Richardson, A. E.: Complex forms of soil organic phosphorus–A major component of soil phosphorus, Environmental Science & Technology, 49, 13238-13245, 10.1021/acs.est.5b02948, 2015b.

McLaren, T. I., Verel, R., and Frossard, E.: The structural composition of soil phosphomonoesters as determined by solution $^{31}$P NMR spectroscopy and transverse relaxation ($T_2$) experiments, Geoderma, 345, 31-37, https://doi.org/10.1016/j.geoderma.2019.03.015, 2019.

McLaren, T. I., Smernik, R. J., McLaughlin, M. J., Doolette, A. L., Richardson, A. E., and Frossard, E.: Chapter Two - The chemical nature of soil organic phosphorus: A critical review and global compilation of quantitative data, in: Advances in Agronomy, edited by: Sparks, D. L., Academic Press, 51-124, 2020.

Nebbioso, A., and Piccolo, A.: Basis of a Humeomics Science: Chemical Fractionation and Molecular Characterization of Humic Biosuprastructures, Biomacromolecules, 12, 1187-1199, 10.1021/bm101488e, 2011.

Smernik, R. J., and Dougherty, W. J.: Identification of phytate in phosphorus-31 nuclear magnetic resonance spectra: the need for spiking, Soil Science Society of America Journal, 71, 1045-1050, 10.2136/sssaj2006.0295, 2007.

Spain, A. V., Tibbett, M., Ridd, M., and McLaren, T. I.: Phosphorus dynamics in a tropical forest soil restored after strip mining, Plant and Soil, 427, 105-123, 10.1007/s11104-018-3668-8, 2018.

Turner, B. L.: Soil organic phosphorus in tropical forests: an assessment of the NaOH–EDTA extraction procedure for quantitative analysis by solution $^{31}$P NMR spectroscopy, European Journal of Soil Science, 59, 453-466, 10.1111/j.1365-2389.2007.00994.x, 2008.

Turner, B. L., Cheesman, A. W., Godage, H. Y., Riley, A. M., and Potter, B. V.: Determination of *neo*- and D-*chiro*-inositol hexakisphosphate in soils by solution $^{31}$P NMR spectroscopy, Environ Sci Technol, 46, 4994-5002, 10.1021/es204446z, 2012.

Turner, B. L.: Isolation of inositol hexakisphosphate from soils by alkaline extraction and hypobromite oxidation, in: Inositol Phosphates: Methods and Protocols, edited by: Miller, G. J., Springer US, New York, NY, 39-46, 2020.

Vestergren, J., Vincent, A. G., Jansson, M., Persson, P., Ilstedt, U., Gröbner, G., Giesler, R., and Schleucher, J.: High-resolution characterization of organic phosphorus in soil extracts using 2D $^1$H–$^{31}$P NMR correlation spectroscopy, Environmental Science & Technology, 46, 3950-3956, 10.1021/es204016h, 2012.

Vincent, A. G., Vestergren, J., Gröbner, G., Persson, P., Schleucher, J., and Giesler, R.: Soil organic phosphorus transformations in a boreal forest chronosequence, Plant and Soil, 367, 149-162, 10.1007/s11104-013-1731-z, 2013.